# Discovery of the major 15–30 nt mammalian small RNAs, their biogenesis and function

Hejin Lai[1,3], Ning Feng [1,3] & Qiwei Zhai [1,2] ✉

Small RNAs (sRNAs) within 15-30 nt such as miRNA, tsRNA, srRNA with 3′-OH have been identified. However, whether these sRNAs are the major 15-30 nt sRNAs is still unknown. Here we show about 90% mammalian sRNAs within 15-30 nt end with 2′,3′-cyclic phosphate (3′-cP). TANT-seq was developed to simultaneously profile sRNAs with 3′-cP (sRNA-cPs) and sRNA-OHs, and huge amount of sRNA-cPs were detected. Surprisingly, sRNA-cPs and sRNA-OHs usually have distinct sequences. The data from TANT-seq were validated by a novel method termed TE-qPCR, and Northern blot. Furthermore, we found that Angiogenin and RNase 4 contribute to the biogenesis of sRNA-cPs. Moreover, much more sRNA-cPs than sRNA-OHs bind to Ago2, and can regulate gene expression. Particularly, snR-2-cP regulates Bcl2 by targeting to its 3′UTR dependent on Ago2, and subsequently regulates apoptosis. In addition, sRNA-cPs can guide the cleavage of target RNAs in Ago2 complex as miRNAs without the requirement of 3′-cP. Our discovery greatly expands the repertoire of mammalian sRNAs, and provides strategies and powerful tools towards further investigation of sRNA-cPs.

High-throughput sequencing has substantially facilitated the discovery of mammalian sRNAs such as miRNA, piRNA, tRNA-derived small RNA (tsRNA) and small rDNA-derived RNA (srRNA)[1–4]. miRNAs within the size of 15−30 nt have been considered as key regulators of biological processes[5]. However, construction of the cDNA libraries of 15−30 nt sRNAs including miRNA for sequencing is usually based on ligation of adapter to 3′-OH of sRNAs[6]. Modifications of RNA have been identified in many different types of small RNAs[7–9], and whether there is any modification at 3′-end severely hampering the ligation is still largely unknown. Furthermore, whether sRNAs with 3′-OH (sRNA-OHs) are the major 15−30 nt sRNAs is yet to be elucidated.

miRNAs play important gene-regulatory roles in animals by guiding Argonaute (AGO) proteins to target sites in the 3′UTR of mRNAs[10,11]. The mammalian genome encodes four AGO proteins (Ago1–4), and Ago2 is the most highly expressed and the only AGO protein able to cleave a target that is fully complementary to the guide strand of the miRNA[10,12]. Interestingly, srRNA, tsRNA, snoRNA, snRNA, small intron RNA (sinR-NAs), small mature mRNA (smRNA), piRNA, small repeat-derived RNA (srpRNA) and small miscRNA (smcRNA) have also been found in Ago2 complex[13–16]. For example, tsRNA CU1276 associated with AGO proteins modulates proliferation and DNA damage response by targeting 3′UTR of *RPA1*[17]. Small RNAs from snoRNA ACA45 existed in Ago2 complex have miRNA-like functions by targeting 3′UTR of *CDC2L6*[16]. However, whether mammalian sRNAs in Ago2 complex have 3′-end modification is still unclear, and whether sRNA-OHs are the major 15−30 nt sRNAs in Ago2 complex remains unknown.

## Results

### The 15−30 nt mouse and human sRNAs mainly end with 3′-cP

To investigate the 3′-end of sRNAs, 15−30 nt sRNAs from mouse liver were analyzed by ligation assay for sRNA-OHs. Only about 10% of the sRNAs were ligated, whereas the synthetic sRNA-OHs can be almost completely ligated under the same condition (Fig. 1a). To further confirm this observation, RNA polyadenylation assay was performed. Nearly 90% of the sRNAs can't be polyadenylated, meanwhile the synthetic sRNA-OHs can be completely polyadenylated (Fig. 1b). These

[1]CAS Key Laboratory of Nutrition, Metabolism and Food Safety, Shanghai Institute of Nutrition and Health, University of Chinese Academy of Sciences, Chinese Academy of Sciences, Shanghai, China. [2]School of Life Science and Technology, ShanghaiTech University, Shanghai, China. [3]These authors contributed equally: Hejin Lai, Ning Feng. ✉e-mail: qwzhai@sibs.ac.cn

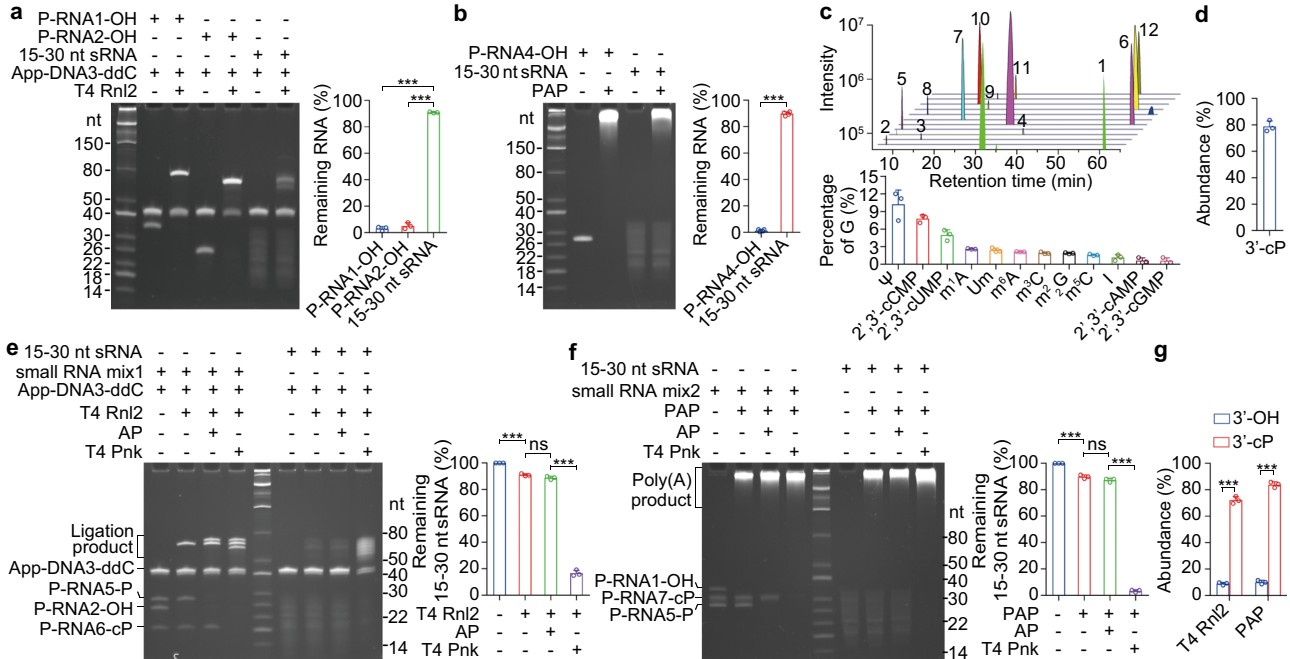

**Fig. 1 | The 15–30 nt sRNAs from mouse liver mainly end with 3′-cP. a, b** Only a small fraction of the 15–30 nt sRNAs from mouse liver can be ligated to the adapter App-DNA3-ddC by RNA ligase T4 Rnl2 (**a**, *n* = 3) or polyadenylated by Poly(A) polymerase (PAP) (**b**, *n* = 4). **c** LC–MS/MS analysis of modified nucleosides and nucleotides from an enzymatic digestion of the 15–30 nt sRNAs. Numbered peaks are 1, m$^1$A; 2, 2′,3′-cUMP; 3, 2′,3′-cGMP; 4, 2′,3′-cAMP; 5, 2′,3′-cCMP; 6, m$^6$A; 7, m$^3$C; 8, ψ; 9, I; 10, m$^5$C; 11, Um; 12, m$^2_2$G *n* = 3. **d** The abundance of the 15–30 nt sRNAs with 3′-cP when analyzed by LC–MS/MS *n* = 3. **e, f** The 3′-end of the 15–30 nt sRNAs is mainly cP when analyzed by ligation assay (**e**) or polyadenylation assay (**f**) *n* = 3. **g** The abundance of the 15–30 nt sRNAs with 3′-OH and 3′-cP when analyzed in (**e, f**) *n* = 3. Data are presented as mean ± SD, statistical significance was determined by two-tailed Student's *t*-test. *$P < 0.05$; **$P < 0.01$; ***$P < 0.001$. ns not significant. Exact *P* values can be found in Source Data Fig. 1. Source data are provided as a Source data file.

results show that only a small proportion of 15–30 nt sRNAs from mouse liver ends with 3′-OH, and suggest that the sRNAs contain modifications, which may block the ligation or polyadenylation at 3′-end of sRNAs.

Based on the LC–MS/MS method to simultaneously quantify 40 different types of nucleosides we reported previously[18], a powerful method to simultaneously quantify more nucleosides and nucleotides was developed to analyze the 15–30 nt sRNAs. Numerous RNA modifications were detected, such as ψ, 2′,3′-cCMP, 2′,3′-cUMP, m$^1$A, Um, m$^6$A, m$^3$C, m$^2_2$G, m$^5$C, I, 2′,3′-cAMP and 2′,3′-cGMP (Fig. 1c and Supplementary Fig. 1a–d). The specificity and accuracy to detect 2′,3′-cNMP were confirmed by using synthetic sRNA-cPs, which were confirmed by polyadenylation assay (Supplementary Fig. 1e, f). Surprisingly, nearly 80% of the 15–30 nt sRNAs contain 3′-cP (Fig. 1d). Similarly, about 80% of the sRNAs from mouse spleen, mouse and human hepatoma cells, Hepa 1–6 and Hep G2, are also with 3′-cP (sRNA-cPs) (Supplementary Fig. 1g, h). These results demonstrate that both mouse and human 15–30 nt sRNAs mainly end with 3′-cP.

To further confirm the above observation, the sRNAs were treated with T4 polynucleotide kinase (T4 Pnk) to remove 3′-phosphate (3′-P) and 3′-cP or alkaline phosphatase (AP) to remove 3′-P, and then analyzed by ligation assay or polyadenylation assay. Treatment with AP only had a slight effect, but treatment with T4 Pnk led to almost completely ligation or polyadenylation of the sRNAs (Fig. 1e, f). These results further confirmed that the 15–30 nt sRNAs mainly end with 3′-cP (Fig. 1g). Similarly, for the sRNAs from mouse spleen, brain, white adipose tissue, Hepa 1–6, AML12, NIH/3T3, Hep G2 and 293 T cells, treatment with AP only had a slight effect, but treatment with T4 Pnk led to sufficient polyadenylation (Supplementary Fig. 1i–k).

All these data demonstrate that both mouse and human 15–30 nt sRNAs mainly end with 3′-cP, and about 90% of 15–30 nt mammalian sRNAs can be categorized as either sRNA-cPs or sRNA-OHs.

## TANT-seq reveals landscape of the 15–30 nt sRNAs, and shows that the sRNA-OHs and sRNA-cPs usually have distinct sequences

To systematically reveal the major mammalian 15–30 nt sRNAs, strategies for sRNA library construction were developed and validated (Supplementary Fig. 2a, b), and further confirmed by PAGE analysis and qPCR (Supplementary Fig. 2c–f). Finally, T4 Rnl2/AP/NaIO₄/T4 Pnk (3′ phosphatase minus)/RtcB-based sRNA-seq (TANT-seq) was used for library construction and high-throughput sequencing to simultaneously profile sRNA-OHs and sRNA-cPs (Fig. 2a). The TANT-seq data show that the abundance of spike-ins with 3′-OH or 3′-cP was exactly as expected (Supplementary Fig. 3a), indicating no significant cross-contamination between 3′-OH and 3′-cP sublibraries. Additionally, the TANT-seq data demonstrated high reproducibility (Supplementary Fig. 3b). By combining the TANT-seq data with ligation efficiency, the relative abundance of sRNA-OHs and sRNA-cPs is about 11–13% and 87–89% respectively in mouse liver, Hepa 1–6 and Hep G2 cells (Fig. 2b and Supplementary Fig. 3c, e).

The sRNAs can be classified into over 10 different biotype classes (Fig. 2c). Interestingly, high abundance of srRNA, tsRNA, sinRNA, small lncRNA (slncRNA), snRNA, small genome-derived RNA (sgmRNA), smRNA, srpRNA, snoRNA, smcRNA and piRNA with 3′-cP in mouse liver, Hepa 1–6 and Hep G2 cells were observed (Fig. 2c and Supplementary Fig. 3d, f).

The relative abundance and length distribution of each biotype class in 3′-OH and 3′-cP sRNA sublibraries is shown in Fig. 2d, e and Supplementary Fig. 3g, h. About 5–10% sRNAs in 3′-OH sublibrary from mouse liver, Hepa 1–6 and Hep G2 cells matched to miRNAs, and less than 0.4% of sRNAs in 3′-cP sublibrary matched to miRNAs (Fig. 2d). The distribution of abundance and unique reads indicates that each biotype class has a lot of high-abundance unique sequences (Supplementary Fig. 3i–k). In addition, the nomenclature and detailed

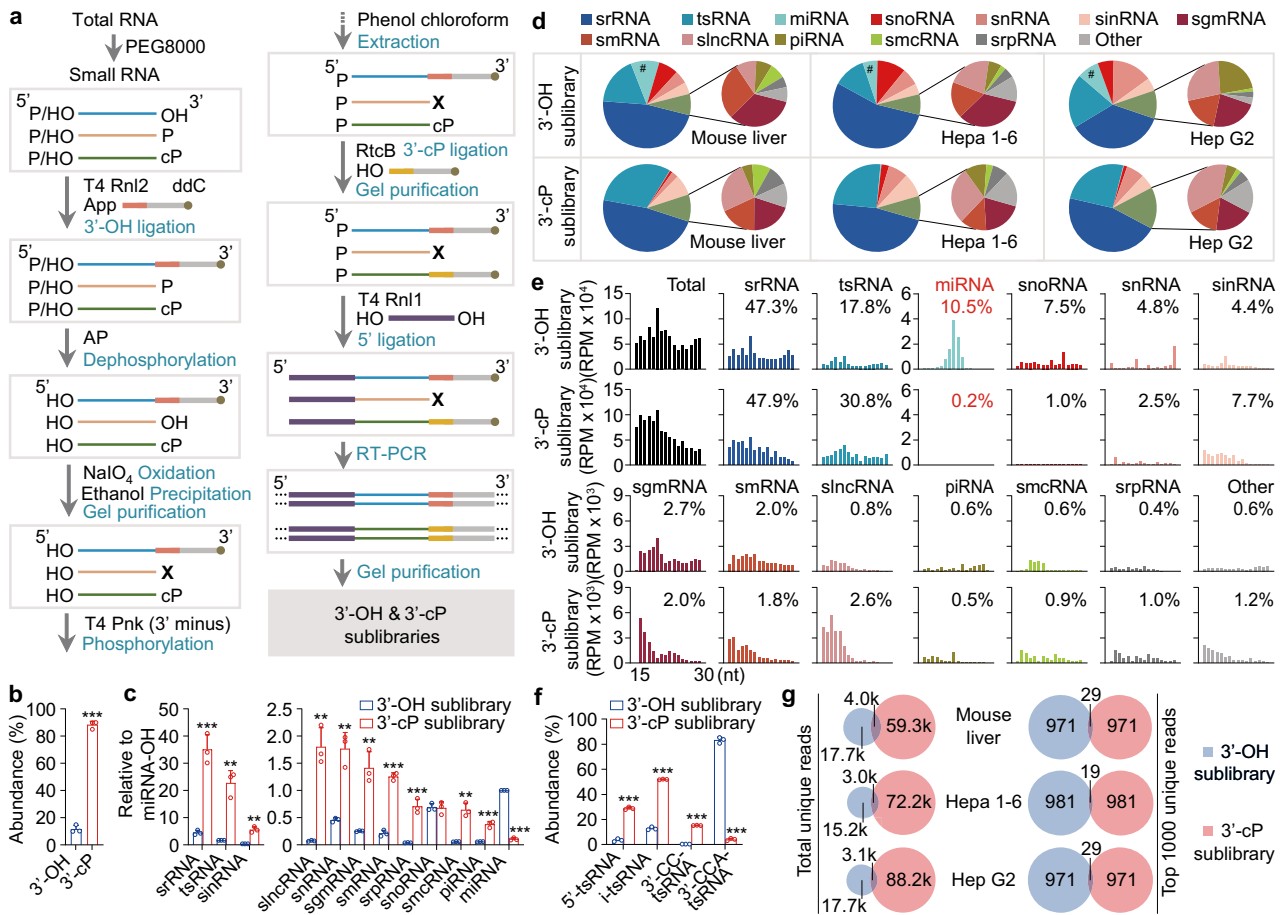

**Fig. 2 | TANT-seq reveals many abundant 15–30 nt mammalian sRNA biotype classes with 3′-cP and 3′-OH. a** Schematic of the workflow for constructing a TANT-seq library to simultaneously profile sRNAs with 3′-OH and 3′-cP. **b** The relative abundance of 15–30 nt sRNAs with 3′-OH or 3′-cP from mouse liver when detected by TANT-seq $n = 3$. **c** TANT-seq reveals many abundant sRNA biotype classes with 3′-OH or 3′-cP in mouse liver $n = 3$. **d** Proportion of sRNA categories detected by TANT-seq. #, indicates a dramatic change between 3′-OH and 3′-cP sublibraries. **e** Length

distribution and abundance of 15–30 nt sRNAs in mouse liver. **f** The percentage of tsRNA classified into each subgroups in 3′-OH and 3′-cP sublibraries is significantly different $n = 3$. **g** The total or top 1000 unique sRNAs between 3′-OH and 3′-cP sublibraries only have a little overlap. Data are presented as mean ± SD, statistical significance was determined by two-tailed Student's $t$-test. *$P < 0.05$; **$P < 0.01$; ***$P < 0.001$. Exact $P$ values can be found in Source Data Fig. 2. Source data are provided as a Source data file.

information of these sRNA-OHs and sRNA-cPs in each biotype class were shown in Supplementary Data 3 to 6.

Although the length distribution of tsRNA in 3′-OH and 3′-cP sublibraries is similar (Fig. 2e and Supplementary Fig. 3g, h), the percentage of different types of tsRNAs is obviously different (Fig. 2f and Supplementary Fig. 3l, m). Similarly, the distribution of tsRNA reads in 3′-OH and 3′-cP sublibraries on a length scale is also obviously different (Supplementary Fig. 4a). Consistently, most srRNA reads in 3′-OH and 3′-cP sublibraries are derived from different loci (Supplementary Fig. 4b–g).

To further reveal the difference between sRNA-OHs and sRNA-cPs, the overlap of unique sequences was analyzed. The unique sRNAs in 3′-OH and 3′-cP sublibraries were pronouncedly different in mouse liver, Hepa 1–6 and Hep G2 cells, and only less than 3% overlap for the top 1000 abundant sRNAs (Fig. 2g). Moreover, the unique sRNAs in each biotype class also only show a little overlap in 3′-OH and 3′-cP sublibraries (Supplementary Fig. 5a–c). The overlap analysis illustrates that sRNA-OHs and sRNA-cPs usually have distinct sequences.

It has been reported that RNA modification may partially interfere with reverse transcription and lead to biased sequencing results, and AlkB has been used to demethylate sRNA in order to improve high-throughput sequencing[19–21]. We confirmed that the AlkB used in this study can dramatically reduce the m[1]A modification in the indicated

purified tRNAs without detectable RNase activity (Supplementary Fig. 6a, b). Then we treated mouse liver sRNAs with AlkB for TANT-seq, and the method using AlkB-treated sRNAs for TANT-seq is termed ATANT-seq. As shown in Supplementary Fig. 6c–e, ATANT-seq reveals overall similar but slightly different sRNA expression pattern compared to TANT-seq. tsRNA-OH was slightly but significantly increased after AlkB treatment (Supplementary Fig. 6f), and the distribution of tsRNA-OH or tsRNA-cP from TNAT-seq and ATANT-seq on a length scale was similar but slightly different. These observations are similar with previous reports[21,22].

These data show that we successfully established an innovative technology termed TANT-seq to simultaneously detect sRNA-OHs and sRNA-cPs, and reveal the landscape of both mouse and human 15–30 nt sRNA-OHs and sRNA-cPs usually with distinct sequences.

## Development of TE-qPCR and validation of the TANT-seq data

To verify the sRNA-cPs detected by TANT-seq, a novel triplex-3′-end qPCR (TE-qPCR) method was developed to simultaneously detect sRNAs with 3′-OH, 3′-P (sRNA-Ps) and 3′-cP. Briefly, the total RNA was treated with vehicle, AP or T4 Pnk, then polyadenylated, reverse transcribed and detected by qPCR (Fig. 3a), and then sRNA-OHs, sRNA-Ps and sRNA-cPs can be quantified. The TE-qPCR was validated by synthetic sRNA-OHs, sRNA-Ps and sRNA-cPs (Fig. 3b).

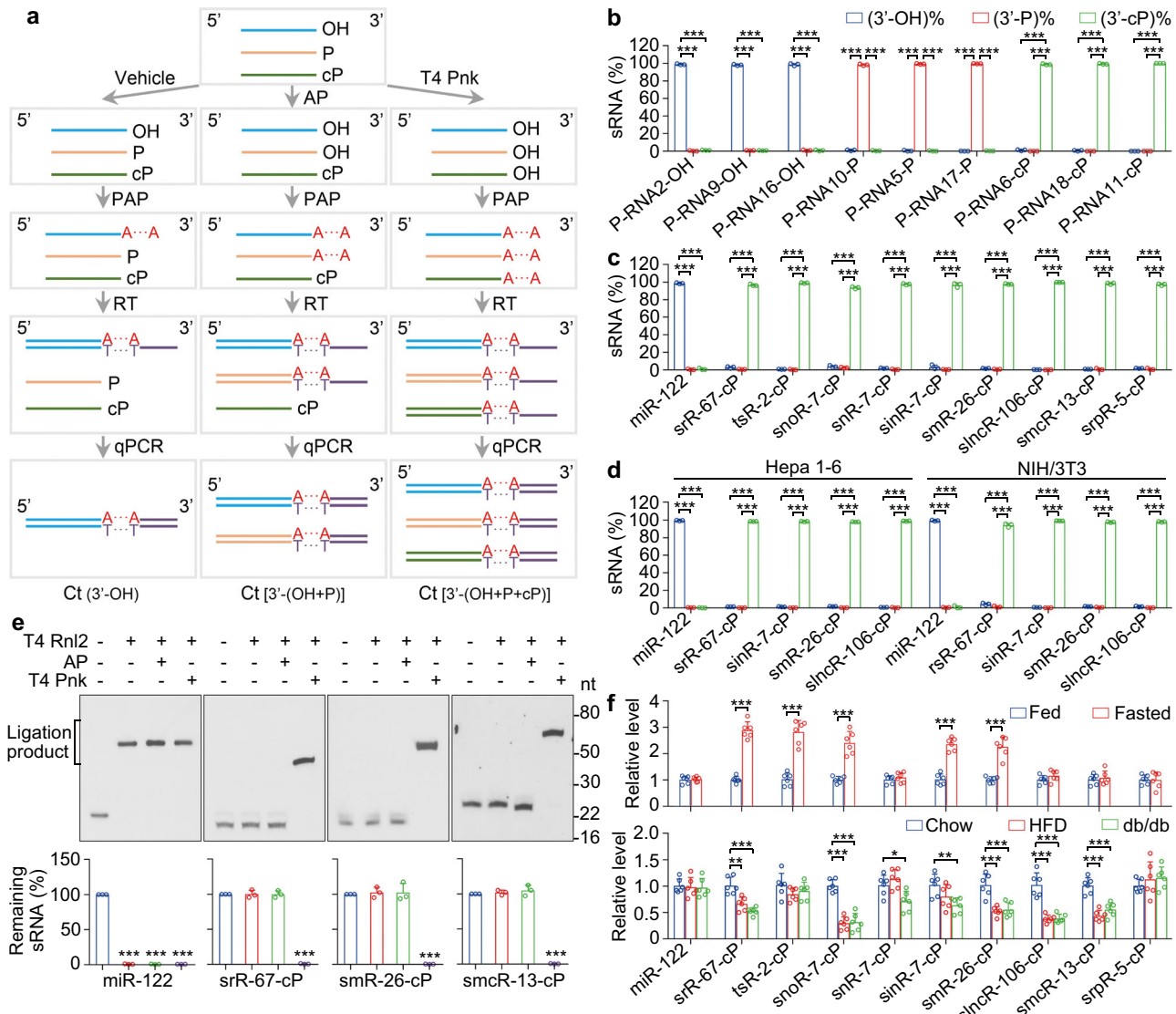

**Fig. 3 | Validation of the TANT-seq data by TE-qPCR and Northern blot.**
**a** Schematic representation of TE-qPCR to detect sRNAs with 3'-OH, 3'-P or 3'-cP.
**b** The indicated 9 synthetic sRNAs were mixed and detected by TE-qPCR $n = 3$.
**c**, **d** The indicated sRNAs in mouse liver (**c**), Hepa 1–6 and NIH/3T3 cells (**d**) were detected by TE-qPCR $n = 3$. **e** The indicated sRNAs in mouse liver were detected by Northern blot after treatment with or without AP or T4 Pnk and ligation with or without T4 Rnl2, and the remaining sRNAs were quantified $n = 3$. **f** The indicated sRNAs in liver of wild-type mice fed, fasted or fed with high-fat diet, or in liver of *db/db* mice were detected by TE-qPCR $n = 6$. Data are presented as mean ± SD. Statistical significance was determined by two-tailed Student's *t*-test. *$P < 0.05$; **$P < 0.01$; ***$P < 0.001$. Exact $P$ values can be found in Source Data Fig. 3. Source data are provided as a Source data file.

The data from TE-qPCR of mouse liver, Hepa 1–6 and Hep G2 samples show that the miRNAs almost 100% end with 3'-OH, and the indicated sRNA-cPs obtained from TANT-seq indeed end with 3'-cP (Fig. 3c, d and Supplementary Fig. 7a, b). Moreover, some sRNA-cPs obtained from TANT-seq were also detected in various tissues and cell lines (Fig. 3d and Supplementary Fig. 7a), suggesting that sRNA-cPs are universally expressed.

The expression and size of the indicated sRNAs obtained from TANT-seq in mouse liver were further confirmed by Northern blot (Fig. 3e and Supplementary Fig. 7c, d). Moreover, miRNAs without any treatment can be ligated or polyadenylated, but only after treatment with T4 Pnk but not AP, the sRNA-cPs can be ligated or polyadenylated (Fig. 3e and Supplementary Fig. 7c, d). These data provide additional evidence to confirm the expression and size of sRNA-cPs obtained from TANT-seq.

The expression of specific sRNA-cPs in different physiological and pathological conditions was detected by TE-qPCR. As shown in Fig. 3f,

rsR-67-cP, snoR-7-cP, sinR-7-cP and smR-26-cP upregulated in liver of fasted mice were significantly decreased in liver of high-fat diet mice or *db/db* mice. These results show that some sRNA-cPs are correlated with physiological and pathological changes.

## Angiogenin and RNase 4 contribute to the biogenesis of sRNA-cPs

Nucleotide enrichment analysis of the sRNAs from TANT-seq revealed strong 3'-terminal enrichment of pyrimidine base and moderate 5'-terminal purine base enrichment of sRNA-cPs (Fig. 4a and Supplementary Fig. 8c, d). For sRNA-OHs, only significant enrichment of 3'-terminal nucleotides was observed in some specific biotype classes such as tsRNA, snoRNA, snRNA, sgmRNA, piRNA and smcRNA (Supplementary Fig. 8).

RNase A has been reported to cleave after pyrimidine base with a preference before purine base to produce 3'-cP[23,24]. As expected, digestion of mouse liver total RNA with RNase A led to a drastic

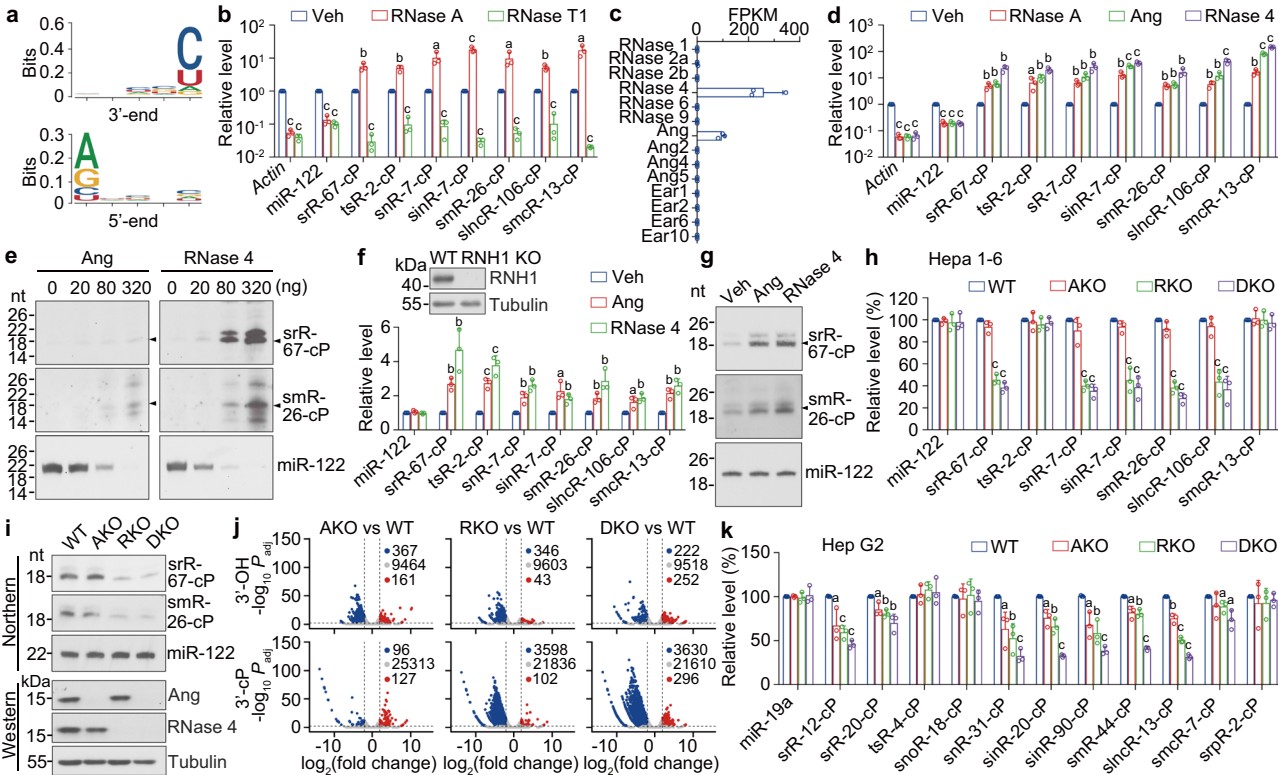

**Fig. 4 | The biogenesis of sRNA-cPs partially depends on Ang and RNase 4.**
**a** Nucleotide enrichment graph for 15–30 nt sRNA-cPs from mouse liver. **b** The relative RNA level detected by TE-qPCR in mouse liver total RNA after digestion with RNase A or RNase T1. Veh, vehicle $n = 3$. **c** The expression level of RNase A family members in mouse liver $n = 3$. **d** The relative RNA level detected by TE-qPCR in mouse liver total RNA after digestion with RNase A, Ang or RNase 4 $n = 3$. **e** Northern blot analysis of the sRNA level in mouse liver total RNA after digestion with Ang or RNase 4. Arrowhead indicates the band with expected level. **f, g** Transfection with Ang or RNase 4 markedly increased the level of indicated sRNA-cPs in *RNH1* KO Hepa 1–6 cells when detected by TE-qPCR (**f**) or Northern blot (**g**) $n = 3$. **h, i** The level of indicated sRNAs in WT, *Ang* KO (AKO), *RNase 4* KO (RKO)

and *Ang/RNase 4* double KO (DKO) Hepa 1–6 cells pretreated with RNase inhibitor when detected by TE-qPCR (**h**) or Northern blot (**i**) $n = 3$. **j** Volcano plots showing differentially expressed 15–30 nt sRNAs with 3'-OH or 3'-cP from WT, AKO, RKO or DKO Hepa 1–6 cells when detected by TANT-seq. **k** The level of indicated sRNAs in WT, AKO, RKO and DKO Hep G2 cells pretreated with RNase inhibitor when detected by TE-qPCR $n = 3$. Data are presented as mean ± SD. Statistical significance was determined by two-tailed Student's *t*-test, except in (**j**) adjusted *P* value was calculated by Benjamini-Yekutieli method. **a** $P < 0.05$; **b** $P < 0.01$; **c** $P < 0.001$. Exact *P* values can be found in Source Data Fig. 4. Source data are provided as a Source data file.

decrease of *Actin* and miR-122, and a significant increase of the indicated sRNA-cPs (Fig. 4b). Meanwhile, digestion with RNase T1, cleaving after G to produce 3'-cP[25], led to an obvious decrease of *Actin* and indicated sRNAs (Fig. 4b). Interestingly, only Angiogenin (Ang) and RNase 4 in RNase A family are highly expressed in mouse liver (Fig. 4c). Digestion with Ang or RNase 4 led to a dramatic decrease of *Actin* and miR-122, but a significant increase of the sRNA-cPs (Fig. 4d and Supplementary Fig. 9a, b). Northern blot analysis further confirmed that digestion with Ang or RNase 4 dose-dependently increased the levels of the indicated sRNA-cPs (Fig. 4e and Supplementary Fig. 9c). These data suggest that Ang and RNase 4 are responsible for the biogenesis of sRNA-cPs in mouse liver.

Transfection of Ang or RNase 4 into Hepa 1–6 cells led to a moderate increase of sRNA-cPs when detected by TE-qPCR (Supplementary Fig. 9d). RNH1 can bind to Ang or RNase 4 and inhibit their activities[26]. Therefore, we established a stable *RNH1* knockout (KO) Hepa 1–6 cell line, and validated by sequencing and immunoblot (Supplementary Fig. 9e and Fig. 4f). Transfection with Ang or RNase 4 led to a significant increase of the sRNA-cPs in *RNH1* KO Hepa 1–6 cells (Fig. 4f, g and Supplementary Fig. 9f). These data show that Ang or RNase 4 contributes to the production of some sRNA-cPs in vitro or in living cells.

To investigate whether Ang or RNase 4 is required for the biogenesis of sRNA-cPs, we established stable *Ang* KO (AKO), *RNase 4* KO (RKO) and *Ang/RNase 4* double KO (DKO) Hepa 1–6 cell lines, and

validated by sequencing and immunoblot (Supplementary Fig. 9g and Fig. 4i). Some sRNA-cPs in RKO and DKO Hepa 1–6 cells were moderately decreased (Supplementary Fig. 9h). Because RNases in serum for cell culture can enter into cells[27], Hepa 1–6 cells were pretreated with RNH1 to block exogenous RNase activity. The levels of srR-67-cP, snR-7-cP, sinR-7-cP, smR-26-cP and slncR-106-cP were markedly decreased in RKO and DKO Hepa 1–6 cells pretreated with RNH1 (Fig. 4h, i and Supplementary Fig. 9j, k). These data demonstrate that RNase 4 is required for the biogenesis of some sRNA-cPs.

In addition, the level of 15–30 nt total sRNAs but not sRNAs with 3'-OH in RKO and DKO Hepa 1–6 cells pretreated with RNH1 was significantly decreased when detected by Northern blot (Supplementary Fig. 9i), suggesting the significant decrease of sRNA-cPs in RKO and DKO Hepa 1–6 cells pretreated with RNH1.

To systematically investigate the effects of AKO, RKO and DKO on 15–30 nt sRNA levels in Hepa 1–6 cells, TANT-seq was performed. 367, 346, 222 sRNA-OHs were significantly decreased in AKO, RKO and DKO cells, meanwhile 96, 3598, 3630 sRNA-cPs were significantly decreased respectively (Fig. 4j). The detailed TANT-seq data of 15–30 nt sRNAs from WT, AKO, RKO and DKO Hepa 1–6 cells were shown in Supplementary Data 3 to 4. These data show that RNase 4 has an important effect on the biogenesis of sRNA-cPs in mice.

To further confirm the effect of Ang and RNase 4 on biogenesis of sRNA-cPs in human, we established stable AKO, RKO and DKO Hep G2 cell lines, and validated by sequencing (Supplementary Fig. 9l). The

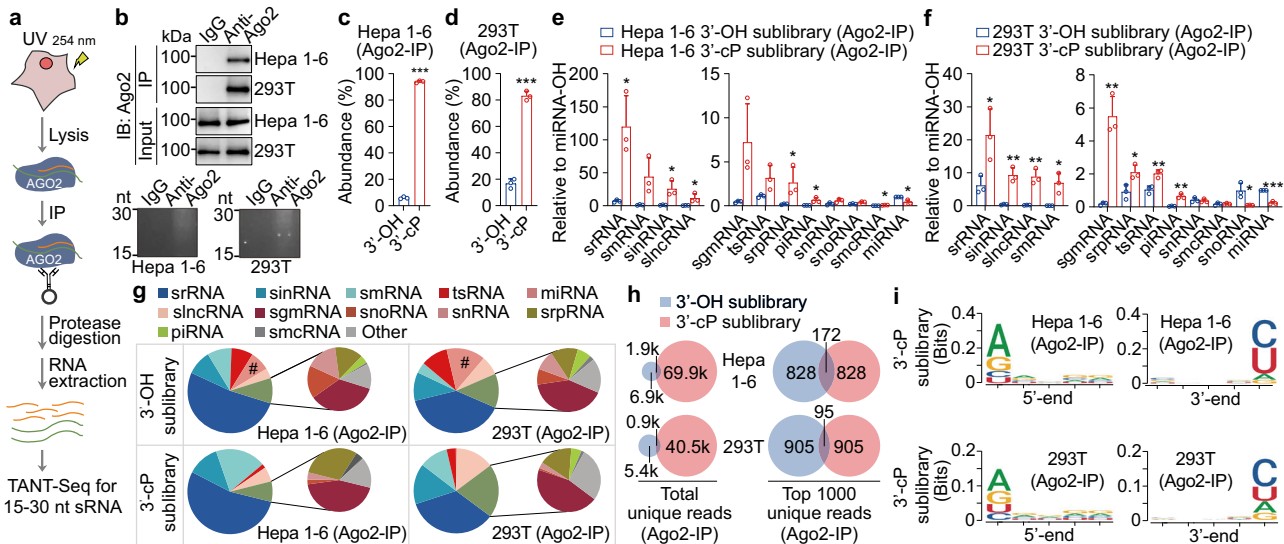

**Fig. 5 | Much more sRNA-cPs than sRNA-OHs bind to Ago2. a** Schematic of immunoprecipitating sRNAs in Ago2 complex for TANT-seq.
**b** Immunoprecipitation (IP) of Ago2 complex from the lysates of Hepa 1–6 cells and 293 T cells was confirmed by immunoblot, and 15–30 nt sRNAs extracted from Ago2 complex were confirmed by denatured PAGE analysis. **c, d** The relative abundance of 15–30 nt Ago2-binding sRNAs with 3′-OH or 3′-cP from Hepa 1–6 (**c**) and 293 T (**d**) cells detected by TANT-seq *n* = 3. **e, f** TANT-seq reveals many abundant sRNA biotype classes with 3′-OH or 3′-cP in Ago2 complex from Hepa 1–6 (**e**) and 293 T (**f**) cells *n* = 3. **g** Proportion of sRNA categories in Ago2 complex from Hepa 1–6 and 293 T cells *n* = 3. **h** The total or top 1000 unique sRNAs between 3′-OH and 3′-cP sublibraries in Ago2-IP samples only have a little overlap in Hepa 1–6 and 293 T cells. **i** Nucleotide enrichment graph for 15–30 nt Ago2-binding sRNA-cPs in Hepa 1–6 and 293 T cells. Data are presented as mean ± SD. Statistical significance was determined by two-tailed Student's *t*-test. \**P* < 0.05; \*\**P* < 0.01; \*\*\**P* < 0.001. Exact *P* values can be found in Source Data Fig. 5. Source data are provided as a Source data file.

levels of srR-12-cP, srR-20-cP, snR-31-cP, sinR-20-cP, sinR-90-cP, smR-44 and slncR-13 were markedly decreased in AKO, RKO and DKO Hep G2 cells pretreated with RNH1 (Fig. 4k). These data show that both Ang and RNase 4 are required for the biogenesis of some mammalian sRNA-cPs.

## Much more sRNA-cPs than sRNA-OHs are in Ago2 complex

It has been reported that Ago2 binds to srRNA, tsRNA, snoRNA, snRNA, sinRNA, smRNA, piRNA, srpRNA and smcRNA besides miRNA[13,14], and tsRNA can physically associate with AGO proteins to repress mRNA transcripts in a sequence-specific manner[17]. To investigate whether sRNA-cPs can bind to Ago2 and regulate gene expression as miRNAs, 15–30 nt sRNAs in Ago2 complex from Hepa 1–6 and 293 T cells were immunoprecipitated and isolated for TANT-seq (Fig. 5a, b). The TANT-seq data of 15–30 nt sRNAs in Ago2 complex were highly reproducible (Supplementary Fig. 10a, b), and the relative abundance of sRNA-OHs and sRNA-cPs is about 5.9% and 94.1%, 16.8% and 83.2% respectively in Hepa 1–6 and 293 T cells (Fig. 5c, d). srRNA-cP, smRNA-cP, sinRNA-cP, slncRNA-cP, sgmRNA-cP, tsRNA-cP and srpRNA-cP have a higher abundance than miRNA in both Hepa 1–6 and 293 T cells (Fig. 5e, f). The relative abundance of each biotype class in 3′-OH and 3′-cP sRNA sublibraries is shown in Fig. 5g, and 7.6% and 14.9% of sRNAs in 3′-OH sublibrary from Hepa 1–6 and 293 T cells matched to miRNA, and only 0.19% and 0.40% of sRNAs in 3′-cP sublibrary matched to miRNA respectively (Fig. 5g). The total and top 1000 unique sRNAs in 3′-OH and 3′-cP sublibraries were pronouncedly different in Ago2 complex (Fig. 5h). Length distribution of the sRNA-OHs or sRNA-cPs in Hepa 1–6 and 293 T cells is shown in Supplementary Fig. 10c, d. The distribution of abundance and unique reads indicates that each biotype class in Ago2 complex has many high-abundance unique sequences (Supplementary Fig. 10e). Nucleotide enrichment analysis of the sRNAs in Ago2 complex also revealed 3′-terminal enrichment of pyrimidine base and 5′-terminal purine base enrichment of sRNA-cPs (Fig. 5i and Supplementary Fig. 10f, g). In addition, the detailed information of top 5000 sRNA-OHs and sRNA-cPs were shown in Supplementary Data 7 to 10.

These data demonstrate that much more sRNA-cPs than sRNA-OHs bind to Ago2, and sRNA-cPs and sRNA-OHs in Ago2 complex usually have distinct sequences.

## sRNA-cPs in Ago2 complex can regulate gene expression and related biological function

To further investigate the potential biological function of sRNA-cPs in Ago2 complex, 35 and 5 sRNA-cPs in and not in Ago2 complex respectively were selected for a three-step screening using luciferase assay. 107 predicted target 3′UTRs of the total 40 sRNA-cPs were selected and cloned into 5 tandem 3′UTR plasmids (Fig. 6a Step 1). After transfection of the tandem 3′UTR plasmids with antisense sRNAs for luciferase assay, 22 antisense sRNAs corresponding to 63 3′UTRs significantly upregulated the luciferase activity over 2 folds in Hepa 1–6 cells (Fig. 6a Step 1 and Supplementary Fig. 11a). Then 63 luciferase plasmids containing a single 3′UTR for each one, were constructed and transfected with the corresponding 22 antisense sRNAs. As shown in Fig. 6a Step 2 and Supplementary Fig. 11b, 20 antisense sRNAs corresponding to 25 3′UTRs significantly upregulated the luciferase activity over 2 folds. Subsequently, the 25 luciferase plasmids were transfected with the corresponding 20 sRNA mimics. As shown in Fig. 6a Step 3 and Supplementary Fig. 11c, 16 sRNA mimics corresponding to 18 3′UTRs significantly downregulated the luciferase activity. Then the 18 luciferase plasmids were mutated at the predicted recognition site and transfected with the corresponding 16 antisense sRNAs or sRNA mimics. As shown in Fig. 6b, mutation of each 3′UTR almost completely blocked the effect of antisense sRNAs or sRNA mimics.

To further confirm the effect of sRNA-cPs on gene expression, 16 antisense sRNAs or sRNA mimics were directly transfected into Hepa 1–6 cells. As shown in Fig. 6c, 10 antisense sRNAs or sRNA mimics significantly upregulated or downregulated 11 mRNAs respectively, and one mRNA at almost undetectable level was not shown. The effect of the 10 antisense sRNAs or sRNA mimics were all significantly dependent on Ago2 (Fig. 6d and Supplementary Fig. 11d, e). The effect of antisense sRNAs or sRNA mimics on corresponding protein levels were further investigated. The protein levels of Nr3c2, Bcl2, Twf1 and

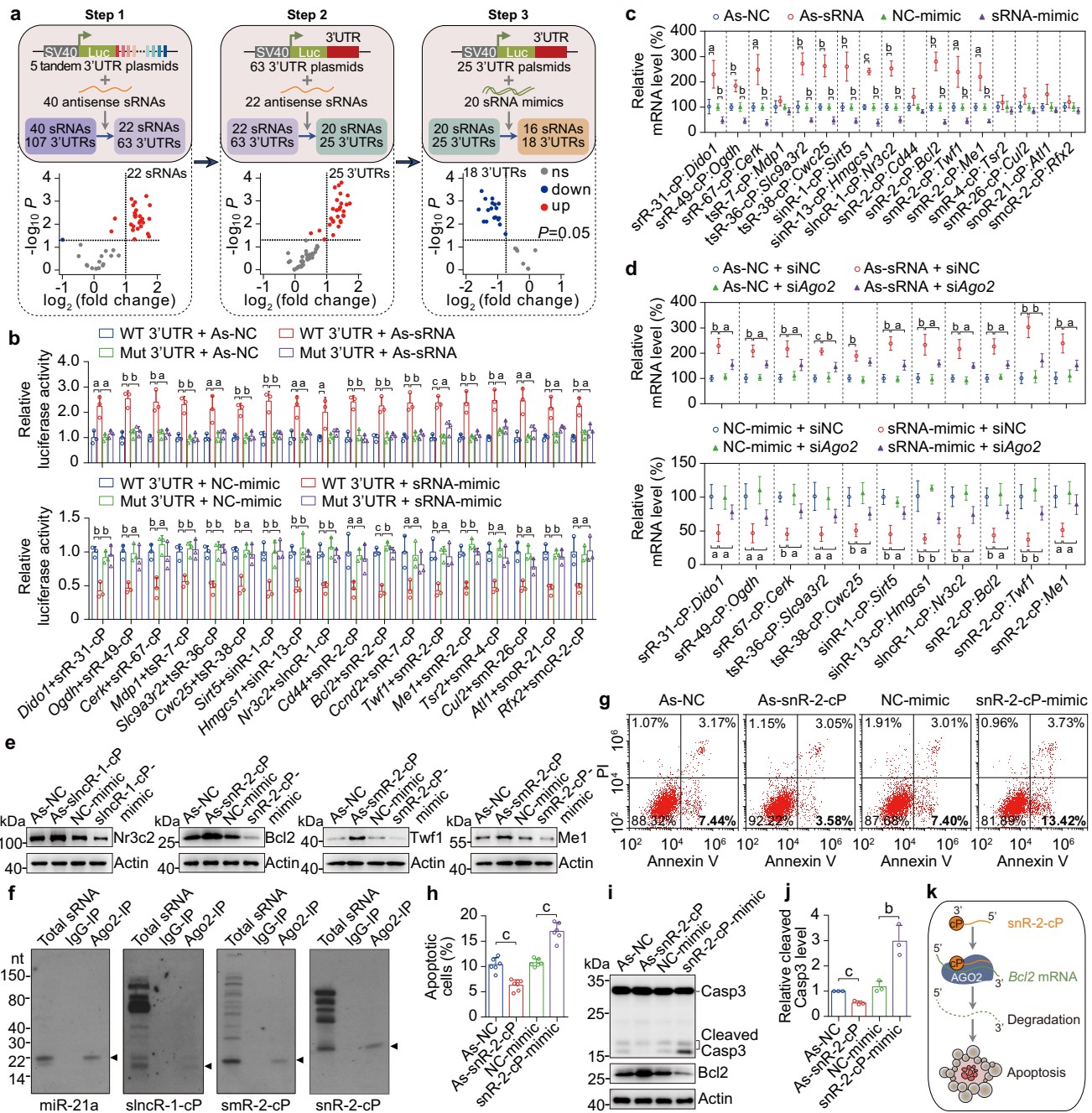

**Fig. 6 | sRNA-cPs can regulate gene expression and related biological function.**
**a** Schematic of a three-step screening using luciferase assay to identify functional
sRNA-cPs and their target 3′UTRs (upper panel), and the data of luciferase assay for
each step (lower panel) $n = 3$. **b** Relative luciferase activity after transfection with the
indicated wild-type or mutant 3′UTR luciferase plasmids and the indicated
antisense sRNAs or sRNA mimics in Hepa 1–6 cells $n = 3$. **c** The mRNA levels
detected by qPCR after transfection with the indicated antisense sRNAs or sRNA
mimics in Hepa 1–6 cells $n = 3$. **d** The mRNA levels after transfection with the
indicated antisense sRNAs or sRNA mimics with or without si*Ago2* in Hepa 1–6 cells
$n = 3$. **e** The protein levels detected by immunoblot after transfection with the
indicated antisense sRNAs or sRNA mimics $n = 3$. **f** The indicated endogenous sRNA-
cPs and miR-21a were detected by Northern blot in total sRNAs or RNAs immuno-
precipitated by IgG or Anti-Ago2 antibody from Hepa 1–6 cells. Arrowhead

indicates the band with expected size. **g** The effect of snR-2-cP on apoptosis
detected by Annexin V-FITC/PI staining. Hepa 1–6 cells transfected with As-snR-2-cP
or snR-2-cP mimic were treated with TNF-α and cycloheximide and analyzed by
Annexin V-FITC/PI staining. **h** Quantification of apoptotic cells in Fig. 6g ($n = 6$).
**i** The effect of snR-2-cP on Caspase-3 (Casp3) cleavage. Hepa 1–6 cells transfected
with As-snR-2-cP or snR-2-cP mimic were treated with TNF-α and cycloheximide and
analyzed by immunoblot. **j** Quantification of the cleaved Casp3 level in Fig. 6i $n = 3$.
**k** Schematic diagram of apoptosis regulated by snR-2-cP via targeting to *Bcl2* 3′UTR
dependent on Ago2 to decrease Bcl2. Data are presented as mean ± SD. Statistical
significance was determined by two-tailed Student's *t*-test. **a** $P < 0.05$; **b** $P < 0.01$;
**c** $P < 0.001$. Exact $P$ values can be found in Source Data Fig. 6. Source data are
provided as a Source data file.

Me1 were markedly regulated by slncR-1-cP, snR-2-cP and smR-2-cP as
indicated (Fig. 6e and Supplementary Fig. 11f). The presence of endo-
genous slncR-1-cP, snR-2-cP and smR-2-cP in extracted sRNAs or
immunoprecipitated Ago2 complex from Hepa 1–6 cells was

confirmed by Northern blot and quantified (Fig. 6f and Supplementary
Fig. 11g).

Bcl2 is well-known for its key role in regulating apoptosis[28]. Since
Bcl2 is regulated by snR-2-cP via targeting to its 3′UTR dependent on

Ago2 (Fig. 6b–d), the effect of snR-2-cP on apoptosis was investigated. As expected, inhibition of snR-2-cP by antisense sRNA markedly blocked apoptosis induced by TNF-α and cycloheximide, and over-expression of snR-2-cP by sRNA mimic significantly enhanced apoptosis when detected by Annexin V-FITC/PI staining and immunoblot of cleaved Caspase-3 (Fig. 6g–j).

These data indicate that some sRNA-cPs can bind to Ago2 like miRNAs, and regulate gene expression and related biological function such as apoptosis (Fig. 6k).

### sRNA-cPs can guide Ago2 cleavage activity as miRNAs without the requirement of 3′-cP

To further investigate whether sRNA-cPs can function in Ago2 complex as miRNAs to cleave target RNA, we first aligned the indicated sRNAs with the synthetic target RNAs and predicted cleavage sites (Fig. 7a–d). Then, we co-transfected a plasmid expressing Myc-Ago2 and the indicated sRNA mimics in 293 T cells, and Myc-Ago2 loaded with indicated sRNA mimics was immunoprecipitated from 293 T cells and used for in vitro cleavage assay. The expression of Myc-Ago2 and the immunoprecipitated Myc-Ago2 were confirmed by immunoblot (Fig. 7a–d). As expected, Myc-Ago2 loaded with snR-2-cP, slncR-1-cP or smR-2-cP mimics significantly cleaved corresponding *Bcl2*, *Nr3C2* or *Twf1* 3′UTR fragments in a time-dependent manner (Fig. 7a–d). The 3′ products from cleavage assay were cloned and sequenced, and the sequences of 3′ products are exactly from the cleavage of target RNA between the nucleotides complementary to nucleotides 10 and 11 of the sRNA-cPs (Fig. 7b–d), which is completely consistent with miRNA as reported previously[29,30]. Except the expected 3′ product, spontaneous uncleaved full-length target RNAs were also cloned due to the insufficient cleavage, low ligation efficiency to the 5′ biotin-labeled end, or the presence of 5′ unlabeled target RNA at a low concentration.

To exclude the non-specific cleavage of Myc-Ago2 loaded with the indicated sRNA mimics, we tested non-specific RNA as the potential target. As shown in Fig. 7e, Myc-Ago2 loaded with miR-122 mimic specifically cleaved miR-122 target but not *Bcl2* 3′UTR fragment, and Myc-Ago2 loaded with snR-2-cP mimic specifically cleaved *Bcl2* 3′UTR fragment but not miR-122 target.

To further confirm whether sRNA-cPs can function in Ago2 complex as miRNAs to cleave target RNA, Myc-Ago2 was immunoprecipitated from 293 T cells and mixed with sRNA mimics for in vitro cleavage assay. As shown in Fig. 7f–i, snR-2-cP, slncR-1-cP and smR-2-cP mimics guided the cleavage of *Bcl2*, *Nr3C2* and *Twf1* 3′UTR fragments respectively in a time-dependent manner as miR-122 guided the cleavage of miR-122 target. These data show that sRNA-cPs can be loaded into Ago2 complex both in vivo and in vitro to guide the cleavage of target RNAs.

To investigate whether sRNA-cPs function in Ago2 complex depends on 5′-P and 3′-cP, snR-2 and smR-2 with 5′-OH, 5′-P, 3′-OH or 3′-cP were prepared. As shown in Fig. 7j, Ago2 complex guided by OH-snR-2-OH and OH-snR-2-cP, or P-snR-2-OH and P-snR-2-cP have similar cleavage activity for *Bcl2* 3′UTR fragment, which suggests 3′-cP is not required for the function of sRNA-cP in Ago2 complex. Similarly, 3′-cP is also not required for smR-2-cP to guide the cleavage of *Twf1* 3′UTR fragment in Ago2 complex (Fig. 7k). However, 5′-P is required for sRNA-cP to sufficiently function in Ago2 complex to cleave target RNA (Fig. 7j, k), which is consistent with the role of 5′-P in miRNA as described previously[31,32].

These data show that sRNA-cPs can specifically guide the cleavage of target RNAs in Ago2 complex as miRNAs without the requirement of 3′-cP.

## Discussion

Here we show that 15–30 nt mammalian sRNAs are mainly sRNA-cPs instead of sRNA-OHs such as miRNA, and only about 10% of 15–30 nt mammalian sRNAs is sRNA-OHs. Consistently, 3′-cP has also been detected in various kinds of RNAs[33–35]. We developed two novel methods, TANT-seq and TE-qPCR, to simultaneously profile and quantify sRNA-OHs and sRNA-cPs. Compared with cP-RNA-seq developed for global identification of sRNA-cPs[36–38], TANT-seq can achieve the unique application to simultaneously profile and quantify sRNA-OHs, sRNA-cPs and their relative abundance, but without measurement of sRNAs with 2′-O-methyl ribose modification as sRNA-cPs[37]. Moreover, cP-RNA-seq has been used to detect gel-purified sRNAs at the size of 30–50 nt or 20–45 nt, which has a high abundance of tsRNAs[36,38]. With TANT-seq, we can detect the low-abundance 15–30 nt sRNAs with high sensitivity directly from sRNAs precipitated by PEG8000, and provide the high-resolution portraits of 15–30 nt sRNAs. sRNA-seq by pretreatment with T4 Pnk or AlkB + T4 Pnk only can obtain the sequences of sRNA-OHs, sRNA-Ps and sRNA-cPs in a mixture without the information for 3′-end[21,39–41]. Pretreatment with AlkB can overcome RNA methylation and improve the quality of sRNA-seq, but AlkB has a risk to cause RNA degradation[41]. In particular, TE-qPCR enables quantification of sRNA-cPs and sRNA-Ps by qPCR. Pretreatment with T4 Pnk followed with qPCR can only quantify sRNAs without the information for 3′-end[36]. With these two novel methods, we revealed and validated over 10 different biotype classes of mouse and human sRNA-OHs and sRNA-cPs, and also provided the nomenclature and related detailed information. Similarly, various biotype classes of sRNA-OHs have also been found in human and mouse samples[42,43]. Interestingly, we found that only about 10.5% of sRNA-OHs in mouse liver is miRNA, which is consistent with the report that about 16.6% of sRNA-OHs in mouse brain is miRNA by SMARTer smRNA-Seq[44]. Based on the importance of sRNA-OHs such as miRNA, the tremendous amount of sRNA-cPs is likely to have crucial roles in many biological processes.

It has been reported that 3′-cP can be generated during cleavage by endonuclease[45], self-cleavage by RNA ribozymes[46], RNA splicing[35], trimming by exonucleases[47] or de novo synthesis by RNA 3′-terminal phosphate cyclase[48]. Here we show that two endonucleases, Ang and RNase 4, contribute to the biogenesis of sRNA-cPs in mammalian cells. The other mechanisms for the biogenesis of sRNA-cPs need to be investigated in the future. The biogenesis of miRNA and piRNA has been shown through Drosha/Dicer-dependent and Zucchini-dependent specific pathways respectively[49], which provides the rationality that the sRNA-OHs and sRNA-cPs usually have distinct sequences.

Interestingly, we found that a lot of sRNAs including sRNA-cPs are in Ago2 complex. Consistently, various types of sRNAs besides miRNA have been found in Ago2 complex[13–16]. Meanwhile, it is possible that many sRNAs identified in this study, including sRNA-cPs, can bind to other Argonaute proteins or other RNA binding proteins to exert their biological functions. Moreover, we found that many sRNA-cPs can regulate mRNA levels in a sequence-specific manner by targeting to the recognition sites in 3′UTRs. Similarly, tsRNA and snoRNA have also been reported to regulate gene expression in mammalian cells by targeting 3′UTR[16,17]. Various types of sRNAs have diverse functions in an Agronaute-dependent manner[50–52]. tsRNAs in human cancer cells and mouse stem cells can enhance mRNA translation and block reverse transcription respectively[53,54]. Transposon-derived small RNAs (small repeat-derived RNA) in Arabidopsis and a 3′UTR-derived small RNA (small mature mRNA) in bacteria can function as small interfering RNAs[55,56]. Interestingly, we further demonstrated that several sRNA-cPs can guide the cleavage of target RNAs in Ago2 complex as miRNAs, which indicates that much more sRNA-cPs can function like miRNAs to exert versatile biological functions, although it still needs to be confirmed in the future.

Our studies greatly expand the repertoire of mammalian sRNAs, and provide an exciting avenue and powerful tools towards further recognition of sRNAs, especially sRNA-cPs, for their discovery, biogenesis and important roles in regulating gene expression and related biological functions.

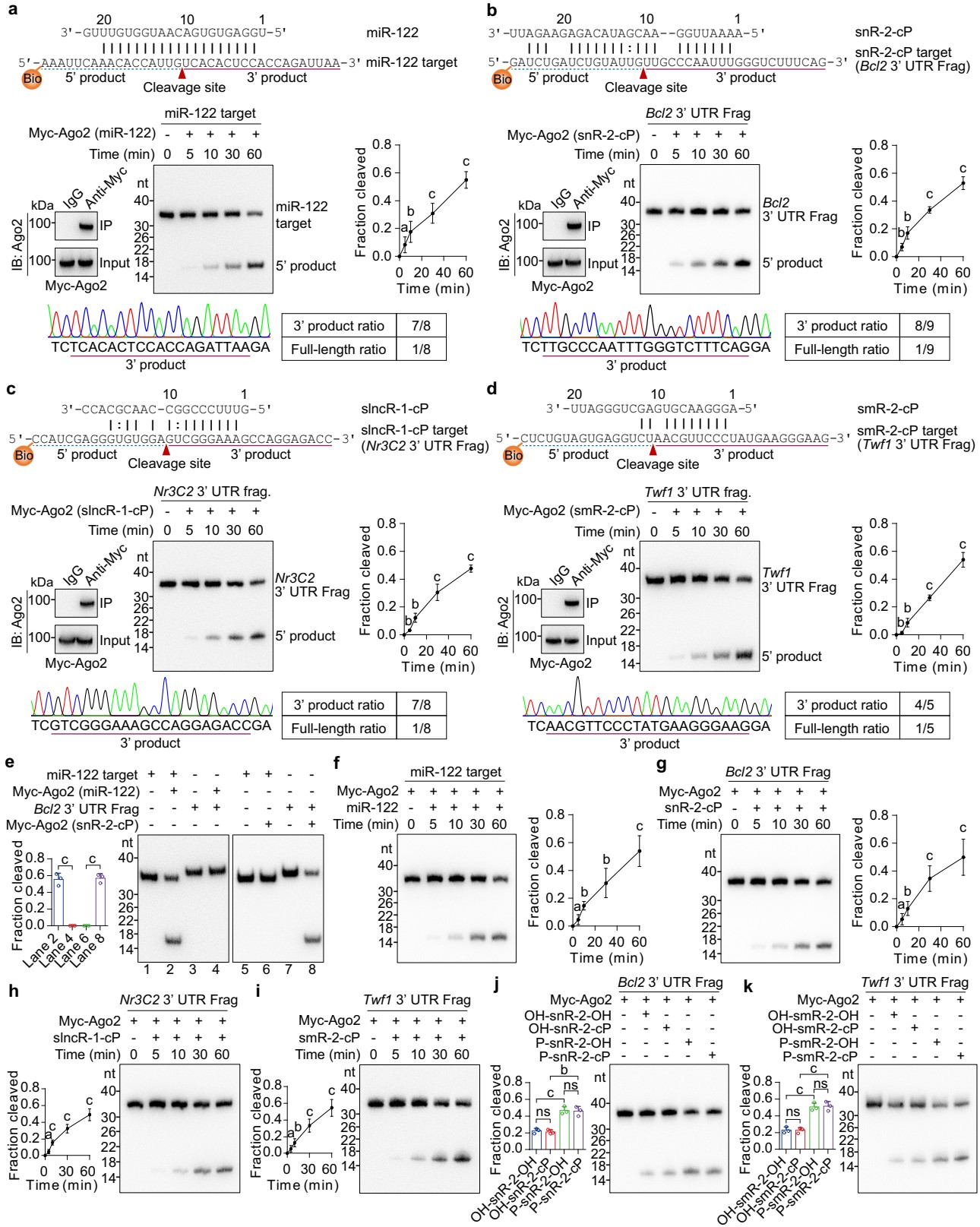

## Methods

### Oligos

DNA and RNA oligos, RNA mimics and siRNAs used in this study were listed in Supplementary Data 1. RNA mimics and siRNAs were obtained by annealing the corresponding sense and antisense sequences.

### Animals

All animals were maintained and used in accordance with the guidelines of the Institutional Animal Care and Use Committee (ethics committee approval no. SIBS-2017-ZQW-1, SINH-2021-ZQW-1, SINH-2022-ZQW-1).

**Fig. 7 | sRNA-cPs can guide the cleavage of target RNAs in Ago2 complex as miRNAs. a–d** In vitro cleavage assay using Myc-Ago2 loaded with miR-122 (**a**), snR-2-cP (**b**), slncR-1-cP (**c**), or smR-2-cP (**d**) immunoprecipitated from 293 T cells. The sequences of sRNAs and the corresponding biotin-labeled target RNA fragments, and the predicted cleavage sites are showed. Myc-Ago2 loaded with indicated sRNA mimics was immunoprecipitated from lysates of 293 T cells transfected with a plasmid expressing Myc-Ago2 and the indicated sRNA mimics, and then validated by immunoblot for in vitro cleavage assay. The representative chromatograms of DNA sequences for 3′ product and 3′ product ratios are also showed. Frag, fragment *n* = 3. **e** In vitro cleavage assay using Myc-Ago2 loaded with miR-122 or snR-2-cP

mimic immunoprecipitated from 293 T cells to confirm the specificity of cleavage *n* = 3. **f–i** Myc-Ago2 immunoprecipitated from 293 T cells transfected with a plasmid expressing Myc-Ago2 was mixed with miR-122 (**f**), snR-2-cP (**g**), slncR-1-cP (**h**), or smR-2-cP mimic (**i**) for in vitro cleavage assay *n* = 3. **j, k** Myc-Ago2 immunoprecipitated from 293 T cells was mixed with snR-2 (**j**) or smR-2 (**k**) containing the indicated 5′ and 3′ ends for in vitro cleavage assay *n* = 3. Data are presented as mean ± SD. Statistical significance was determined by two-tailed Student's *t*-test. **a** *P* < 0.05; **b** *P* < 0.01; **c** *P* < 0.001. ns not significant. Exact *P* values can be found in Source Data Fig. 7. Source data are provided as a Source data file.

To perform the fasting experiment, male C57BL/6 mice at the age of 8 weeks were randomly grouped and maintained in standard cages with or without access to food for 24 h, and water was available *ad libitum* in a room with an ambient temperature of 20–24 °C and humidity of 40–60% on a 12 h light/dark cycle. To study the effect of high-fat diet, male C57BL/6 mice at the age of 8 weeks were randomly assigned to feed chow or high-fat diet with 60 kcal% fat (Research Diets) for 16 weeks. Male *db/db* mice at the age of 16 weeks were used.

Tissues of interest were collected and snap-frozen in liquid nitrogen immediately after resection and stored at −80 °C.

### Cell culture
Hepa 1–6, AML12, NIH/3T3, Hep G2 and 293 T cells, obtained from National Collection of Authenticated Cell Cultures, CAS, China, were maintained in DMEM with 10% fetal bovine serum. The cells with 80–90% confluence were used for total RNA extraction and isolation of sRNAs.

### Denatured PAGE analysis
For denatured PAGE analysis, RNA or DNA oligo samples were mixed with equal volume of 2 × RNA loading buffer (95% formamide, 18 mM EDTA, 0.025% SDS, 0.025% bromophenol blue, 0.025% xylene cyanol). After incubation at 70 °C for 10 min, the samples were immediately placed on ice for 15% denaturing PAGE containing 7 M urea, and the gel was stained by SYBR Gold nucleic acid gel stain (Invitrogen). Gels were visualized and quantified with ImageJ (https://imagej.nih.gov/ij/index.html).

### Gel extraction
The gel containing the sRNAs or DNA oligos was excised, crushed and soaked in 0.3 M NaCl with constant rotation at 4 °C for about 2 h, and the supernatants were precipitated with ethanol-NaAc (3 volumes of absolute ethanol and 0.1 volume of 3 M sodium acetate, pH 5.2). Finally, the sRNAs or DNA oligos were reconstituted in RNase-free water.

### Adenylation of DNA oligos
DNA oligos were adenylated using a 5′ DNA Adenylation Kit (New England Biolabs). After the adenylation reaction, the samples were separated by 15% denatured PAGE, and stained with SYBR Gold nucleic acid gel stain. The adenylated oligos were excised from the gel and extracted.

### Preparation of sRNAs with 5′-P and 3′-cP
Preparation of sRNAs with 3′-cP was performed using Mth RNA ligase in 5′ DNA Adenylation Kit as previously described[57]. Briefly, 100 pmol synthetic sRNA with 5′-OH and 3′-P were incubated in 1 × Adenylation reaction buffer, 50 μM ATP and 5 μM Mth RNA ligase in a total volume of 20 μl at 65 °C for 60 min, followed by heat inactivation at 85 °C for 5 min. Then, 1 μl 10 × CutSmart buffer, 1 U shrimp alkaline phosphatase (New England Biolabs) and RNase-free water were added to a final volume of 30 μl and incubated at 37 °C for 30 min to convert the remaining 3′-P to 3′-OH. After heat inactivation at 65 °C for 20 min, 1 μl 10 × Poly(A) polymerase reaction buffer, 4 μl 10 mM ATP, 10 U

recombinant ribonuclease inhibitor (Takara), 10 U Poly(A) polymerase (New England Biolabs) and RNase-free water were added to a final volume of 40 μl, and incubated at 37 °C for 30 min. Then the samples were separated by 15% denatured PAGE, and the sRNAs with 5′-OH and 3′-cP were excised from the gel and extracted.

To obtain sRNAs with 5′-P and 3′-cP, 300 pmol sRNA with 5′-OH and 3′-cP, 5 μl 10 × T4 Pnk reaction buffer, 5 μl 10 mM ATP, 40 U recombinant ribonuclease inhibitor, 10 U T4 Pnk (3′ phosphatase minus) (New England Biolabs) and RNase-free water were mixed in a total volume of 50 μl, and incubated at 37 °C for 60 min. After heat inactivation at 65 °C for 20 min, the samples were recovered with QIAquick Nucleotide Removal Kit (Qiagen) and dissolved in RNase-free water to obtain sRNAs with 5′-P and 3′-cP.

### Total RNA extraction and isolation of sRNAs
Total RNA was extracted using Trizol reagent (Invitrogen) according to the manufacturer's instructions. High molecular weight RNAs ( > 200 nt) were removed from total RNA by precipitation with 5% polyethylene glycol 8000 and 0.5 M NaCl on ice for 30 min as described previously[58]. sRNAs ( < 200 nt) in the supernatants were precipitated with ethanol-NaAc overnight at −80 °C, and dissolved in RNase-free water and stored at −80 °C. sRNAs ( < 200 nt) were separated by 15% denatured PAGE, and the gel was stained by SYBR Gold nucleic acid gel stain. The 15–30 nt sRNAs were excised from the gel and extracted.

### Ligation assay for sRNA-OHs
Ligation assay for sRNA-OHs was carried out in 20 μl final volume containing 50 ng sRNAs, 20 pmol adenylated DNA oligo, 2 μl 10 × T4 RNA ligase reaction buffer, 4 μl 50% PEG8000, 20 U recombinant ribonuclease inhibitor, 200 U T4 RNA ligase 2, truncated KQ (T4 Rnl2) (New England Biolabs) and RNase-free water. The mixtures were incubated at 16 °C for 16 h, and the ligation products were analyzed by denatured PAGE.

### RNA polyadenylation assay
RNA polyadenylation assay was performed in a volume of 20 μl containing 50 ng RNA samples, 2 μl 10 × Poly(A) polymerase reaction buffer, 2 μl 10 mM ATP, 20 U recombinant ribonuclease inhibitor, 5 U Poly(A) polymerase and RNase-free water, and incubated at 37 °C for 30 min. The polyadenylated products were analyzed by 15% denatured PAGE.

### Analysis of RNA modification by LC−MS/MS
Adenosine (A), uridine (U), cytidine (C), guanosine (G), 2′-O-methyladenosine (Am), 1-methyladenosine (m[1]A), $N^6$-methyladenosine (m[6]A), $N^6$-isopentenyladenosine (i[6]A), 2′-O-methyluridine (Um), 5-methyl-2-thiouridine (m[5]s[2]U), 5-methoxyuridine (mo[5]U), 5-methyluridine (m[5]U), 4-thiouridine (s[4]U), 3′-O-methyluridine (3′-OMeU), 2′-O-methylcytidine (Cm), 5-methylcytidine (m[5]C), $N^4$-acetylcytidine (ac[4]C), 2′-O-methylguanosine (Gm), $N^2$-methylguanosine (m[2]G), 7-methylguanosine (m[7]G), inosine (I), adenosine 2′,3′-cyclic monophosphate (2′,3′-cAMP), uridine 2′,3′-cyclic monophosphate (2′,3′-cUMP), cytidine 2′,3′-cyclic monophosphate (2′,3′-cCMP) and guanosine 2′,3′-cyclic monophosphate (2′,3′-cGMP) were purchased from Sigma-Aldrich. 5,2′-O-dimethyluridine

(m$^5$Um), 3-methyluridine (m$^3$U), 2-thiocytidine (s$^2$C), 5-hydroxymethylcytidine (hm$^5$C), $N^2$-dimethylguanosine (m$^2_2$G), pseudouridine (ψ), 1-methylpseudouridine (m$^1$ψ) and 2′-$O$-methylinosine (Im) were obtained from Berry and Associates Inc. 3′-$O$-methyladenosine (3′-$O$MeA), 2-thiouridine (s$^2$U), 3-methylcytidine (m$^3$C), 3′-$O$-methylcytidine (3′-$O$MeC), 5,2′-$O$-dimethylcytidine (m$^5$Cm), $N^4$-acetyl-2′-$O$-methylcytidine (ac$^4$Cm), 1-methylguanosine (m$^1$G), 3′-$O$-methylguanosine (3′-$O$MeG), $N^2$,$N^2$,7-trimethylguanosine (m$^{2,2,7}$G), 5′-$O$-methylthymidine (5′-$O$MeT) and 3′-$O$-methylinosine (3′-$O$MeI) were purchased from Carbosynth. Guanosine 5′-monophosphate ($^{13}$C$_{10}$, $^{15}$N$_5$) was obtained from Cambridge Isotopes Laboratories. Guanosine 5′-monophosphate ($^{13}$C$_{10}$, $^{15}$N$_5$) at 140 µg/ml was hydrolyzed by calf intestine alkaline phosphatase (Takara) at a final concentration of 1 U/µl at 37 °C for 24 h to obtain guanosine ($^{13}$C$_{10}$, $^{15}$N$_5$). Enzymatic digestion of sRNAs and LC−MS/MS analysis were performed as previously described[18]. Briefly, 100 ng sRNAs were digested with 0.2 U nuclease P1 (Sigma-Aldrich) in 60 µl of 50 mM NH$_4$OAc, pH 5.3 at 50 °C for 3 h and then treated with 0.04 U phosphodiesterase I (USB) for 2 h at 37 °C. Subsequently, the RNA samples were treated with 2 U alkaline phosphatase (Sigma-Aldrich) for 2 h at 37 °C. Proteins were removed by centrifugation through Nanosep 3 K device with Omega membrane (Pall). Then, the samples were lyophilized and stored at −80 °C and were resuspended with 17.5 ng/ml G ($^{13}$C, $^{15}$N) in 100 µl of 2 mM ammonium acetate for LC−MS/MS analysis. Analysis of nucleoside mixtures was performed on an API 4000 Q-TRAP mass spectrometer (Applied Biosystems) with an Agilent 1200 HPLC system and a diode array UV detector (190−400 nm) and equipped with an electrospray ionization source. The calibration curves for LC−MS/MS analysis are shown in Supplementary Data 2.

### Treatment with shrimp alkaline phosphatase or T4 Pnk
50 ng indicated synthetic or isolated sRNAs, 1 µl 10 × CutSmart buffer, 20 U recombinant ribonuclease inhibitor, 1 U shrimp alkaline phosphatase and RNase-free water were mixed in a total volume of 10 µl, and incubated at 37 °C for 30 min. After heat inactivation at 65 °C for 20 min, the samples were used for ligation assay or RNA polyadenylation assay.

50 ng indicated synthetic or isolated sRNAs, 1 µl 10 × T4 Pnk reaction buffer, 20 U recombinant ribonuclease inhibitor, 10 U T4 Pnk (New England Biolabs) and RNase-free water were mixed in a total volume of 10 µl, and incubated at 37 °C for 30 min. After heat inactivation at 65 °C for 20 min, the samples were used for ligation assay or RNA polyadenylation assay.

### RtcB ligation assay
15−30 nt sRNAs used for RtcB ligation assay were treated with T4 Pnk (3′ phosphatase minus) to phosphorylate the 5′-OH of sRNAs according to the manufacturer's instructions. After heat inactivation, the samples were extracted with phenol-chloroform, precipitated with ethanol-NaAc, and dissolved in RNase-free water. RtcB ligation assay was carried out in 20 µl final volume containing 50 ng sRNAs, 20 pmol adapter (OH-RNA8-OH), 2 µl 10 × RtcB reaction buffer, 2 µl 10 mM MnCl$_2$, 1 µl 10 mM GTP, 15 pmol RtcB ligase (for ligation of 3′-adapter to sRNAs with 3′-P or 3′-cP, New England Biolabs), 20 U recombinant ribonuclease inhibitor and RNase-free water. The mixtures were incubated at 37 °C for 2 h, and the ligation products were analyzed by 15% denatured PAGE.

### Crosslinking immunoprecipitation and RNA extraction
Cells cultured in 10 cm dishes at a confluence of about 80% were washed twice with ice-cold PBS. After aspiration of the PBS, the cells were treated with UV irradiation at 254 nm (400 mJ/cm$^2$) on ice in Stratalinker 1800 (Stratagene). Subsequently, 500 µl lysis buffer (20 mM Tris−HCl pH 7.5, 137 mM NaCl, 1% Nonidet P-40, 2 mM EDTA, 0.5 mM DTT, 100 U/ml recombinant ribonuclease inhibitor, 1 × proteinase inhibitor cocktail (Roche)) was added and incubated on ice for

10 min. The lysates were centrifuged at 12000 g for 10 min at 4 °C to collect supernatant.

For immunoprecipitation, 20 µl of protein A/G magnetic beads slurry (Thermo Fisher Scientific) was washed with lysis buffer and incubated with 2 µg anti-Ago2 antibody (Abcam) or normal rabbit IgG antibody (CST) in 200 µl lysis buffer for 4 h at 4 °C. The antibody-bound beads were washed with lysis buffer for three times, and incubated with the supernatant of cell lysates for 4 h at 4 °C. Then the beads were washed with lysis buffer for three times, and subsequently used for immunoblot or RNA extraction.

For RNA extraction, the beads were resuspended in 400 µl 100 mM Tris−HCl pH 7.5, 50 mM NaCl, 10 mM EDTA containing 10 mg/ml protease K (Millipore) and incubated for 30 min at 37 °C. Finally, the Ago2-binding RNA was obtained by phenol-chloroform extraction, precipitation with ethanol-NaAc, and reconstitution in RNase-free water.

### Small RNA library preparation and high-throughput sequencing
To construct the library for high-throughput sequencing, 1 µg sRNAs (< 200 nt) in a volume of 9 µl mixed with 1 µl spike-in mixture (P-RNA2-OH, P-RNA10-P and P-RNA6-cP, 8 nM each; P-RNA9-OH, P-RNA5-P and P-RNA18-cP, 0.8 nM each; P-RNA16-OH, P-RNA17-P and P-RNA11-cP, 0.08 nM each), or 100 ng RNA obtained from crosslinking immunoprecipitation in a volume of 10 µl, were incubated at 70 °C for 5 min and immediately placed on ice. Then, the samples were mixed with 20 pmol App-DNA12-ddC containing the barcode for 3′-OH sublibrary, 4 µl 10 × T4 RNA ligase reaction buffer, 8 µl 50% PEG8000, 400 U T4 RNA ligase 2, truncated KQ, 40 U recombinant ribonuclease inhibitor and RNase-free water in a 40 µl reaction volume, and incubated at 16 °C for 16 h. After heat inactivation at 70 °C for 5 min, 1 µl 10 × CutSmart buffer, 20 U recombinant ribonuclease inhibitor, 2.5 U shrimp alkaline phosphatase and RNase-free water were added to a final volume of 50 µl and incubated at 37 °C for 30 min. Subsequently, 20 µl 100 mM NaIO$_4$, 40 U recombinant ribonuclease inhibitor and RNase-free water were added to a final volume of 200 µl and incubated at 0 °C for 40 min in the dark to convert the 2′ and 3′-OH at 3′-end into dialdehydes[59]. After precipitation with ethanol-NaAc, the sRNAs at the size of 15-46 nt were purified with 15% denatured PAGE, and treated with T4 Pnk (3′ phosphatase minus) to phosphorylate the 5′-OH of sRNAs, followed by phenol-chloroform extraction and precipitation with ethanol-NaAc. After dissolved in 10 µl RNase-free water, 10 pmol OH-RNA13-ddC containing the barcode for 3′-cP sublibrary, 2 µl 10 × RtcB reaction buffer, 2 µl 10 mM MnCl$_2$, 1 µl 10 mM GTP, 15 pmol RtcB ligase, 20 U recombinant ribonuclease inhibitor and RNase-free water were added to a final volume of 20 µl, and incubated at 37 °C for 2 h, followed by heat inactivation at 70 °C for 5 min. The 3′-adapter-ligated sRNAs at the size of 31-46 nt were purified with 15% denatured PAGE. The purified sRNAs were mixed with 5 pmol OH-RNA14-OH, 3 µl 10 × T4 RNA ligase reaction buffer, 6 µl 50% PEG8000, 1 µl 10 mM ATP, 30 U T4 RNA ligase 1 (New England Biolabs), and 20 U recombinant ribonuclease inhibitor in a final volume of 30 µl, and incubated at 25 °C for 2 h. Then, 3 pmol OH-DNA15-OH, 200 U SuperScript IV Reverse Transcriptase (Invitrogen), 4 µl 5 × SSIV buffer, 2.5 µl 100 mM DTT, 2.5 µl 10 mM each dNTP, and 20 U recombinant ribonuclease inhibitor and RNase-free water were added to a final volume of 50 µl, and incubated at 50 °C for 1 h, followed by heat inactivation at 80 °C for 10 min. The cDNA was amplified using Phusion High-Fidelity DNA Polymerase (New England Biolabs) with RNA PCR primer and RNA PCR index primers (Supplementary Data 1). The amplified products at the size of 130−160 bp were purified with 7.5% native PAGE, and then subjected to high-throughput sequencing using Xten platform (Illumina). This T4 Rnl2/AP/NaIO$_4$/T4 Pnk (3′ phosphatase minus)/RtcB-based sRNA-seq is termed TANT-seq. Negative control library, 3′-OH library and 3′-OH & 3′-P & 3′-cP library were constructed as the above description, except T4 Rnl2 and RtcB ligase, RtcB ligase, shrimp alkaline phosphatase were replaced with nuclease free water respectively.

For AlkB treatment, 2 μg sRNAs ( < 200 nt) were incubated in 20 μl reaction mixtures containing 2 μl 10 × AlkB Buffer I, 2 μl 10 × AlkB Buffer II, 20 U AlkB (Beyotime), 40 U recombinant ribonuclease inhibitor at 37 °C for 60 min. Then, the mixtures were used for denatured PAGE analysis, or extracted with 500 μl TRIzol reagent to obtained purified AlkB-treated sRNAs. The AlkB-treated sRNAs were subsequently used for TANT-seq. The method using AlkB-treated sRNAs for TANT-seq is termed ATANT-seq.

### Analysis of small RNA sequencing data

Raw reads were first trimmed with fastp (v0.20.0)[60] to remove adapter sequences except the barcode to distinguish 3′-OH and 3′-cP sublibraries. Then low-quality (Phred score <30) reads, and reads shorter than 21 nt or longer than 46 nt were filtered out. The sequences with the barcode GACGTA in App-DNA12-ddC or CTATCG in OH-RNA13-ddC were assigned to 3′-OH sublibrary and 3′-cP sublibrary respectively, and then the barcodes were removed.

Subsequently, the reads were sequentially mapped to the corresponding reference database using ncbi-BLAST + 2.11.0. The CCA sequence was added to the 3′-ends of all tRNAs from GtRNAdb[61] and mitotRNAdb[62] before mapping. The reads were first mapped to spike-ins (Supplementary Data 1) and mouse rRNAs including mmu-5S rRNA (NR_030686.1), mmu-5.8 S rRNA (NR_003280.2), mmu-12S rRNA (NC_005089.1), mmu-16S rRNA (NC_005089.1), mmu-18S rRNA (NR_003278.3), mmu-28S rRNA (NR_003279.1), mmu-45S rRNA (NR_046233.2) and mouse rRNA from Ensembl[63], or human rRNAs including has-5S rRNA (NR_023363.1), has-5.8 S rRNA (NR_145821.1), has-12S rRNA (NC_012920.1), has-16S rRNA (NC_012920.1), has-18S rRNA (NR_146146.1, NR_145820.1), has-28S rRNA (NR_003287.4, NR_146118.1, NR_146154.1, NR_146148.1, NR_145822.1), has-45S rRNA (NR_046235.3, NR_146117.1, NR_146151.1, NR_146144.1) and human rRNA from Ensembl, then sequentially mapped to mouse or human tRNA modified from GtRNAdb with additional 3′-CCA, mitochondrial tRNA modified from mitotRNAdb with additional 3′-CCA, miRNA from miRbase[64] and Ensembl, piRNA from piRNAdb (https://www.pirnadb.org), snRNA, snoRNA, lncRNA, miscRNA and other ncRNAs from Ensembl, intron obtained from the information of exon sequence from Ensembl, mature mRNA, IG gene, TR gene, Pseudogene and TEC from Ensembl, repeat sequence (RepeatMasker, GRCm38/mm10 or GRCh38/hg38) from UCSC Genome Browser (http://genome.ucsc.edu), genome (GRCm38/mm10 or GRCh38/hg38) and mitochondrial genome (NC_005089.1 or NC_012920.1), and the perfectly matched reads were assigned to the mapped biotype classes. Then the unmapped reads were sequentially mapped to the above database again allowing mismatches + gaps ≤2, and assigned to the mapped biotype classes. Unmapped sequences were discarded.

The sequences mapped to repeat sequence assigned to pseudogene in Dfam[65], were manually picked and assigned to pseudogene. Some high abundant sRNAs for each biotype class were manually checked with NCBI nucleotide-nucleotide BLAST. In mouse samples, the sequences mapped to repeat sequence matched to small nucleolar RNA, C/D box 118 (Snord118) (NR_028566.3) or Gm25313 (XR_004935972.1) were manually picked and assigned to snoRNA or snRNA respectively. The sequences mapped to mRNA sequence perfectly matched to Snord14c (NR_028276.2) were manually picked and assigned to snoRNA. In human samples, the sequences mapped to repeat sequence perfectly matched to U5A small nuclear 1 (RNU5A-1) (NR_002756.2), and U5B small nuclear 1 (RNU5B-1) (NR_002757.3) were manually picked and assigned to snRNA. The sequences mapped to repeat sequence matched to small nucleolar RNA C/D box 118 (SNORD118) (NR_033294.1) were manually picked and assigned to snoRNA.

The reads mapped to rRNAs and tRNAs were classified into srRNA and tsRNA respectively. The reads mapped to intron, lncRNA, genome, mature mRNA, repeat sequence and miscRNA were termed as small intron RNA (sinRNA), small lncRNA (slncRNA), small genome-derived RNA (sgmRNA), small mature mRNA (smRNA), small repeat-derived RNA (srpRNA), small miscRNA (smcRNA) respectively. A small RNA nomenclature was established for sRNAs with average reads more than 2 according to its biotype class, description of matched sequence, start position, length and 3′-modification, meanwhile a simplified small RNA nomenclature was also established according to its biotype class, rank of abundance and 3′-modification. tsRNAs were further classified into 5′-tsRNA (tRNA-derived sRNA containing 5′-terminal), 3′-CCA-tsRNA (tRNA-derived sRNA containing the additional CCA at 3′-terminal), 3′-CC-tsRNA (tRNA-derived sRNA containing the additional CC at 3′-terminal) and internal-tsRNA.

For replicate analysis, independent biological replicates of TANT-seq experiments were compared by computing the Pearson correlation coefficient by GraphPad Prism 7.0 between any two replicate experiments.

The relative abundance of 15–30 nt sRNAs or each biotype class of sRNA with 3′-OH and 3′-cP was calculated as the following. The ligation efficiency of 15–30 nt sRNAs with 3′-OH to 3′-adapter by T4 Rnl2 was calculated with the data shown in Fig. 1e, f considering that 15–30 nt sRNAs with 3′-OH could be completely polyadenylated by Poly(A) polymerase. The ligation efficiency of 15–30 nt sRNAs with 3′-P and 3′-cP to 3′-adapter by RtcB ligase was calculated with the data shown in Supplementary Fig. 1f and Supplementary Fig. 2a. Then the reads of sRNAs with 3′-OH and 3′-cP were adjusted with the ratio of ligation efficiency for T4 Rnl2 and RtcB ligase to calculate the relative abundance of 15–30 nt sRNAs or each biotype class of sRNA with 3′-OH and 3′-cP.

Venn diagrams were generated by VennDiagram package[66].

To obtain the nucleotide enrichment, the nucleotide abundance was calculated from the terminal five nucleotides in 3′-OH or 3′-cP sublibrary. The sequence logos were constructed with the R package ggseqlogo[67].

For differential expression analysis of sRNAs, the fold changes and $P$ values were calculated by edgeR[68], and adjusted $P$ value was calculated by Benjamini-Yekutieli method as described previously[69]. sRNAs with adjusted $P$ value < 0.01 and $\log_2$(fold change) ≥2 were considered as differentially expressed. The sequences with average read ≥2 were shown in the volcano plot.

The distribution of abundance and unique reads for sRNAs was visualized in weighted scatter plots using the function geom_count from the package ggplot2 in R (version 3.3.5; https://ggplot2.tidyverse.org).

The distribution of srRNA or tsRNA reads on a length scale was obtained by an R script to show their expression levels and positional mapping information. srRNA reads were directly mapped to their respective precursors. tsRNA reads were mapped to their respective precursors and then aligned on a length scale from 5′ ends of tRNA.

### Quantitative PCR

Quantitative PCR was performed using FastStart Universal SYBR Green Master (Roche). Universal qPCR reverse primer 1 and the indicated synthetic RNA qPCR forward primers listed in Supplementary Data 1 were used.

### Quantification of sRNAs with 3′-OH, 3′-P or 3′-cP

Schematic representation of triplex-3′-end qPCR (TE-qPCR) to detect sRNAs with 3′-OH, 3′-P or 3′-cP was shown in Fig. 3a. 1 μl spike-in mixture (P-RNA2-OH, P-RNA10-P and P-RNA6-cP, 8 nM each; P-RNA9-OH, P-RNA5-P and P-RNA18-cP, 0.8 nM each; P-RNA16-OH, P-RNA17-P and P-RNA11-cP, 0.08 nM each) or 1 μg total RNA was treated with vehicle, shrimp alkaline phosphatase or T4 Pnk in a 10 μl reaction volume, respectively, followed by heat inactivation at 65 °C for 10 min. Then, 1 μl 10 × Poly(A) polymerase reaction buffer, 2 μl 10 mM ATP, 5 U recombinant ribonuclease inhibitor, 5 U Poly(A) polymerase and RNase-free water were added to a final volume of 20 μl, and incubated at 37 °C for 60 min, followed by heat inactivation at 70 °C for 5 min. Subsequently, 3 pmol universal reverse transcription primer as

indicated in Supplementary Data 1, 200 U SuperScript IV Reverse Transcriptase, 4 µl 5×SSIV buffer, 2.5 µl 100 mM DTT, 2.5 µl 10 mM each dNTP, and 20 U recombinant ribonuclease inhibitor and RNase-free water were added to a final volume of 40 µl, and incubated at 50 °C for 60 min, followed by heat inactivation at 80 °C for 10 min. Quantitative PCR was performed using FastStart Universal SYBR Green Master and the qPCR primers for RNA quantification are shown in Supplementary Data 1 to obtain Ct (3′-OH), Ct [3′-(OH + P)] and Ct [3′-(OH + P + cP)]. The relative abundance of sRNAs with 3′-OH, 3′-P or 3′-cP was calculated according to the following formulas:

$$(3'\text{-cP})\% = \frac{2^{Ct[3'-(OH+P)] - Ct[3'-(OH+P+cP)]} - 1}{2^{Ct[3'-(OH+P)] - Ct[3'-(OH+P+cP)]}} \qquad (1)$$

$$(3'\text{-P})\% = \frac{2^{Ct(3'-OH) - Ct[3'-(OH+P+cP)]} - 1}{2^{Ct(3'-OH) - Ct[3'-(OH+P+cP)]}} - (3'\text{-cP})\% \qquad (2)$$

$$(3'\text{-OH})\% = 100\% - (3'\text{-P})\% - (3'\text{-cP})\% \qquad (3)$$

### tRNA purification and AlkB treatment

In a total 500 µl volume, 1 mg total RNA were hybridized with 5 µl 10 µM biotin-labeled DNA oligos complementary to an indicated tRNA (Supplementary Data 1) in 1×SSC buffer at 50 °C for 16 h. Then 50 µl streptavidin sepharose (GE Healthcare) were added and incubated at room temperature for 30 min, and then the sepharose was washed five times with 1×SSC buffer. The tRNAs were eluted from the sepharose by incubation with 10 mM Tris−HCl, pH 7.6 at 70 °C for 5 min, and then the supernatant was precipitated with ethanol-NaAc, and reconstituted in RNase-free water. For AlkB treatment, 100 ng purified indicated tRNA was incubated in 20 µl reaction mixtures containing 2 µl 10×AlkB Buffer I, 2 µl 10×AlkB Buffer II, 20 U AlkB (Beyotime), 40 U recombinant ribonuclease inhibitor at 37 °C for 60 min. Then, AlkB-treated tRNA was extracted with 500 µl TRIzol reagent for LC−MS/MS analysis.

### Northern blot

Northern blot was performed as previously described with minor modifications. In brief, 20 µg total RNA, 5 µg sRNA (<200 nt), 100−200 ng 15−30 nt sRNAs, or 100−200 ng RNA immunoprecipitated by anti-Ago2 antibody were separated by 15% denatured PAGE and transferred to positively charged Nylon Membranes (Roche). Subsequently, the membranes were air-dried and UV crosslinked, and pre-hybridized with DIG Easy Hyb buffer (Roche). Then, the membranes were incubated overnight with 10 nM 3′-end digoxigenin-labeled oligonucleotides probes as shown in Supplementary Data 1 at 45−55 °C mainly depends on the annealing temperature of each probe. The membranes were washed and blocked, and incubated with Anti-Digoxigenin-AP Fab fragments (Roche) and detected with CSPD ready-to-use reagent (Roche). Quantification of detected sRNAs was performed with ImageJ. The absolute quantification of sRNAs was performed using synthetic sRNAs detected by Northern blot at various concentrations to establish standard curves for each specific sRNA.

### Recombinant protein expression and purification

The Angiogenin (Ang) and RNase 4 cDNA were obtained by RT-PCR from mouse liver total RNA with primers listed in Supplementary Data 1, and inserted into pET28a at the sites of Nde I and Xho I to obtain pET28a-Ang and pET28a-RNase 4. Recombinant Ang and RNase 4 were obtained by IPTG-inducible expression in *Escherichia coli* BL21 Rosetta (DE3) cells and purification with His-Tag Purification Resin (Roche) and ion-exchange column (Hitrap Q HP, GE Healthcare). Purified Ang and RNase 4 were analyzed by SDS-PAGE and Coomassie blue staining, and then dialyzed to storage buffer (20 mM Tris−HCl pH 7.5, 200 mM NaCl and 5% glycerol) and stored at −80 °C.

### Digestion of liver total RNA with RNases

4 µg mouse liver total RNA was treated with 0.5 ng RNase A (Thermo Fisher Scientific), 1 U RNase T1 (Thermo Fisher Scientific), 0.32 µg Ang, 0.08 µg RNase 4 or the indicated amount of Ang or RNase 4 in a final volume of 20 µl at 37 °C for 30 min. Then, the samples were extracted with phenol-chloroform, precipitated with ethanol-NaAc, and dissolved in RNase-free water for TE-qPCR or Northern blot.

### Analysis of the expression level of RNase A family members

RNA-seq data of mouse liver were obtained from the Gene Expression Omnibus (GEO) database (accession code GSE164819[70]. The expression levels were presented using Fragments Per Kilobase of exon model per Million mapped fragments (FPKM).

### CRISPR/Cas9 gene editing

To construct the sgRNA expression plasmids, complementary oligonucleotides indicated in Supplementary Data 1 encoding gRNAs were annealed and cloned into BsmBI site of lentiCRISPR v2 vector (Addgene plasmid 52961) and confirmed by sequencing to obtain lentiCRISPR-mRNH1-sgRNA, lentiCRISPR-mAng-sgRNA1, lentiCRISPR-mRNase 4-sgRNA, lentiCRISPR-hAng-sgRNA1, lentiCRISPR-hRNase 4-sgRNA. To construct the sgRNA expression plasmids px330-mp-mAng-sgRNA2 and px330-mp-hAng-sgRNA2, complementary oligonucleotides indicated in Supplementary Data 1 encoding gRNAs were annealed and cloned into the Bbs I site of px330-mp[71].

The Hepa 1−6 or Hep G2 cells were transfected with lentiCRISPR v2 vector or the constructed plasmids by Lipofectamine 3000 transfection reagent (Thermo Fisher Scientific) according to the manufacturer's instructions. The transfected cells were screened with 2 ng/µl puromycin (Thermo Fisher Scientific) for 48 h. Thereafter, the cells were digested and diluted to obtain single-cell clones. Knockout of the indicated genes was confirmed by immunoblot and/or sequencing to obtain wild-type (WT), *Ang* knockout (AKO) or *RNase 4* knockout (RKO) Hepa 1−6 or Hep G2 cells, and *RNH1* knockout (RNH1 KO) Hepa 1−6 cells.

RKO Hepa 1−6 cells and RKO Hep G2 cells were transfected with px330-mp-mAng-sgRNA2 and px330-mp-hAng-sgRNA2 respectively using Lipofectamine 3000 transfection reagent. After transfection for 48 h, the cells with red fluorescence were sorted with flow cytometry into a 96-well plate containing 1 cell per well. Subsequently, the single-cell clones were collected, and knockout of the indicated genes was confirmed by immunoblot and/or sequencing to obtain *Ang* and *RNase 4* double knockout (DKO) Hepa 1−6 and Hep G2 cells.

### Protein transfection

Wild-type or *RNH1* KO Hepa 1−6 cells were transfected with 2 µg Ang or RNase 4 using Xfect protein transfection reagent (Takara) in 6-well plates according to the manufacturer's protocol. After transfection for 6 h, the cells were washed twice with PBS for subsequent total RNA extraction with Trizol reagent.

### RNase inhibitor pretreatment

Recombinant ribonuclease inhibitor (RNH1) was added into the cell culture media of WT, AKO, RKO or DKO Hepa 1−6 or Hep G2 cells at a final concentration of 2000 U/ml. After incubation for 48 h, the cells were used to extract total RNA with Trizol reagent for TE-qPCR, Northern blot or TANT-seq.

### Immunoblot

The cells were lysed in RIPA lysis buffer containing 50 mM Tris−HCl pH 7.6, 150 mM NaCl, 1% NP-40, 1% sodium deoxycholate, 0.1% SDS. Immunoblot was performed with antibodies against RNH1 (Abclonal, A4079, 1:5000), Angiogenin (Abcam, ab189207, 1:1000), RNase 4 (Abcam, ab200717, 1:1000), Ago2 (Abcam, ab186733, 1:1000), Bcl2 (Abclonal, A19693, 1:1000), Nr3c2 (Abclonal, A3308, 1:1000), Twf1 (Abclonal, A15307, 1:1000), Me1 (Abclonal, A3956, 1:1000), Caspase-3

(CST, 9662, 1:1000), Actin (Sigma-Aldrich, A1978, 1:10000) or Tubulin (Sigma-Aldrich, T6074, 1:10000).

## Detection and quantification of 15−30 nt sRNAs with 3′-(OH + P + cP) or 3′-OH

4 µg total RNA from WT, AKO, RKO or DKO Hepa 1−6 cells pretreated with recombinant ribonuclease inhibitor for 48 h was incubated with or without T4 Pnk to quantitate 15−30 nt sRNAs with 3′-(OH + P+cP) or 3′-OH in 10 µl final volume. Then, 20 pmol 16 nt App-DNA12-ddC, 1 µl of 10 × T4 RNA ligation buffer, 2 µl 50% PEG8000, 20 U recombinant ribonuclease inhibitor, 200 U T4 Rnl2, truncated KQ and RNase-free water were added to a final volume of 20 µl and incubated at 16 °C for 16 h. The ligation products were analyzed by 15% denatured PAGE. Northern blotting was performed with DIG-labeled oligonucleotides probe which was reverse complementary to App-DNA12-ddC as shown in Supplementary Data 1. Ligated sRNAs at the size of 31−46 nt, corresponding to 15−30 nt sRNAs, were analyzed by ImageJ, and normalized to the 5 S rRNA.

## Luciferase reporter assay

Target 3′UTRs of sRNAs were predicted by miRanda[72]. Tandem 3′UTR luciferase reporter plasmids, including pGL3-20-3′UTRs, pGL3-19-3′UTRs, pGL3-22-3′UTRs-a, pGL3-22-3′UTRs-b and pGL3-24-3′UTRs, were constructed by cloning the predicted target 3′UTRs into pGL3-Promoter (Promega) near XbaI site by homologous recombination as shown in Supplementary Data 11.

pGL3-3′UTR plasmid was obtained by insertion of CATATGCCGCGGGATATCCTGCAGGACTAGTCGAATTCCCTAGA into XbaI site of pGL3-Promoter. Luciferase reporter plasmids containing a single 3′UTR were constructed by insertion of a synthetic target 3′UTR sequence at XbaI and EcoRI sites of pGL3-3′UTR as shown in Supplementary Data 11. Mutation of 3′UTR at the seed region is also shown in Supplementary Data 11.

For luciferase reporter assay, Hepa 1−6 cells cultured in a 24-well plate were transfected with the indicated luciferase reporter plasmids, pRL-TK (Promega) and the indicated antisense sRNAs or sRNA mimics using Lipofectamine 3000. After transfection for 48 h, cells were harvested for luciferase reporter assay using Dual-Luciferase Reporter Assay System (Promega). The firefly luciferase activity was normalized to Renilla luciferase activity.

## In vitro cleavage assay using Ago2

The alignment of sRNA sequence and the corresponding biotin-labeled target RNA fragment sequence was predicted by miRanda[72].

Full-length Ago2 cDNA was obtained by RT-PCR using mouse liver total RNA with primers listed in Supplementary Data 1, and inserted into pCMV-Tag 3 A plasmid at the sites of EcoRI and HindIII to obtain pCMV-Myc-Ago2 plasmid. 293 T cells in each 10 cm dish were transfected with 15 µg pCMV-Myc-Ago2 or 10 µg pCMV-Myc-Ago2 mixed with 300 pmol indicated sRNA mimics using Lipofectamine 3000. After 48 h, 500 µl lysis buffer (20 mM Tris−HCl pH 7.5, 137 mM NaCl, 1% Nonidet P-40, 2 mM EDTA, 0.5 mM DTT, 100 U/ml recombinant ribonuclease inhibitor, 1 × proteinase inhibitor cocktail) was added and incubated on ice for 10 min. The lysates were centrifuged at 12000 g for 10 min at 4 °C to collect supernatant. For immunoprecipitation, 20 µl of protein A/G magnetic beads slurry was washed with lysis buffer and incubated with 5 µl Myc Rabbit Polyclonal Antibody (Beyotime) or normal rabbit IgG antibody in 200 µl lysis buffer for 4 h at 4 °C. Then the antibody-bound beads were washed with lysis buffer for three times, and incubated with the supernatant of cell lysates for 4 h at 4 °C. Subsequently the beads were washed with lysis buffer for three times, and Myc-Ago2 complexes were eluted with 50 µl 150 µg/ml c-Myc peptide (Beyotime) according to the manufacturer's instructions.

Ago2 cleavage assay for in vivo loaded sRNAs was essentially performed in 20 µl final volume as described previously[73], containing 5 µl Myc-Ago2 complexes purified from cell lysates co-transfected with the indicated sRNA mimics, 1 nM 5′ end biotin-labeled target RNA (Supplementary Data 1) and 1× cleavage buffer (20 mM Hepes-KOH, pH 7.4, 150 mM KCl, 2 mM MgCl₂, 0.01% Triton X-100, 5% glycerol, 1 mM DTT).

Ago2 cleavage assay for in vitro loaded sRNAs, 5 µl Myc-Ago2 complexes purified from cell lysates without co-transfection with sRNA mimics were mixed with 5 nM indicated sRNA mimics or sRNA in the presence of 1 × cleavage buffer in 10 µl final volume, and incubated at 37 °C for 30 min. Then 1 nM 5′ end biotin-labeled target RNA (Supplementary Data 1) and 1× cleavage buffer at a final concentration were added in a total volume of 20 µl. After incubation for the indicated times at 37 °C, equal volume of 2 × RNA loading buffer was added and heated at 70 °C for 5 min for denatured PAGE analysis. RNA in the gel was transferred to positively charged Nylon Membranes. Subsequently, the membranes were air-dried, UV crosslinked and analyzed with Chemiluminescent Nucleic Acid Detection Module (Thermo Fisher Scientific) according to manufacturer's protocol. Quantification was performed with ImageJ, and the fraction cleaved was calculated as the ratio of the 5′ product density to the density of 5′ product and uncleaved target RNA.

## Cloning and sequencing of the 3′ product from Ago2 cleavage assay

sRNAs after Ago2 cleavage assay were mixed with 180 µl RNase-free water, extracted by phenol-chloroform, precipitated with ethanol-NaAc, and reconstituted in 20 µl RNase-free water. Then 10 µl purified sRNAs were mixed with 5 pmol App-DNA12-ddC, 2 µl 10 × T4 RNA ligase reaction buffer, 4 µl 50% PEG8000, 200 U T4 RNA ligase 2, truncated KQ, 20 U recombinant ribonuclease inhibitor and RNase-free water in a 20 µl reaction volume, and incubated at 16 °C for 16 h. After heat inactivation at 70 °C for 5 min, the samples were extracted with phenol-chloroform and precipitated with ethanol-NaAc. After dissolved in 10 µl RNase-free water, sRNAs were mixed with 5 pmol OH-RNA14-OH, 3 µl 10 × T4 RNA ligase reaction buffer, 6 µl 50% PEG8000, 1 µl 10 mM ATP, 30 U T4 RNA ligase 1, and 20 U recombinant ribonuclease inhibitor in a final volume of 30 µl, and incubated at 25 °C for 2 h. Then, 3 pmol OH-DNA15-OH, 200 U SuperScript IV Reverse Transcriptase, 4 µl 5 × SSIV buffer, 2.5 µl 100 mM DTT, 2.5 µl 10 mM each dNTP, and 20 U recombinant ribonuclease inhibitor and RNase-free water were added to a final volume of 50 µl, and incubated at 50 °C for 1 h, followed by heat inactivation at 80 °C for 10 min. The cDNA was amplified using rTaq DNA Polymerase (Takara) with RNA PCR primer and RNA PCR index primer (Supplementary Data 1), and then purified with 7.5% native PAGE for subsequent TA cloning and sequencing.

## Apoptosis assays

Hepa 1−6 cells were cultured in 12-well plate, and transfected with antisense sRNA or sRNA mimic. After transfection for 36 h, TNF-α (100 ng/ml) and cycloheximide (1 µg/ml) were added and incubated for 4 h. The cells were harvested for immunoblot or flow cytometry analysis with Annexin V-FITC/PI apoptosis detection kit (Vazyme).

## Statistical analysis

All measurements were taken from distinct samples. Except indicated, data are expressed as mean ± SD of at least three independent experiments, and statistical significance was assessed by Student's t-test or edgeR. Except indicated, differences were considered statistically significant at $P < 0.05$. * or a indicates $P < 0.05$, ** or b indicates $P < 0.01$, *** or c indicates $P < 0.001$.

## Reporting summary

Further information on research design is available in the Nature Portfolio Reporting Summary linked to this article.

## Data availability

Raw sequence data have been deposited in the NCBI Sequence Read Archive under BioProject number PRJNA725316. Raw unprocessed LC–MS/MS data have been deposited in Figshare (https://doi.org/10.6084/m9.figshare.23530554.v1). Source data are provided with this paper. All other data and materials are available upon reasonable request. Source data are provided with this paper.

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

## Acknowledgements

We thank past and present members of the Zhai laboratory for helpful discussion; M.H. Yan for help with LC–MS/MS analysis; X. Liu for assistance with bioinformatics analysis; H.B. Zhang's laboratory for providing the plasmid px330-mp. This work was funded by the National Key R&D Program of China to Q.Z. (2018YFA0800603), the National Natural Science Foundation of China to Q.Z. (31630037, 91740103 and 91940306), the Strategic Priority Research Program of the Chinese Academy of Sciences to Q.Z. (XDB19030103), the Frontier Science of Chinese Academy of Sciences Key Research Projects to Q.Z. (QYZDJ-SSW-SMC022) and the Shanghai Leading Talents Program to Q.Z.

## Author contributions

H.L. contributed to the LC–MS/MS analysis and recombinant protein expression and purification. N.F. contributed to bioinformatics analysis. H.L. and N.F. performed all the other experiments. Q.Z., H.L. and N.F. analyzed the data and wrote the manuscript. Q.Z. conceived and designed the project.

## Competing interests

The authors declare no competing interests.
