## [Peer Review File · Nature Communications]

Discovery of the major 15-30 nt mammalian small RNAs, their biogenesis and functionREVIEWER COMMENTS

Reviewer #1 (Remarks to the Author):

The authors adequately addressed most of my comments. I appreciate that. The paper has been very much improved by adding functional data for these RNAs.

However, the reply for this comment below is unsatisfactory.

“2. RNA modification interference: it is widely accepted that at least for tRNA fragments, the interfering modifications make it difficult for quantitative assessments. The authors mentioned this issue in the discussion, and proposes the demethylase treatment as an approach to increase accuracy. Not even trying this approach in this work is not compatible with the conclusions described in the just published work of PANDORA-seq in NCB.”

The authors tried something and gave up in their reply. AlkB treatment and sequencing has been used in many papers now. It is clear that extensive purification of the recombinant protein will be needed to remove contaminating ribonucleases from *E. coli*. For example, the authors can follow the AlkB protein purification steps (better yet use both AlkB and AlkB-D135S mutant, both plasmids can be obtained from Addgene) in this paper: PMID 26214130. Alternatively, they may contact Arraystar, inc. who may provide sequencing services using AlkB.

Reviewer #2 (Remarks to the Author):

In the revised manuscript, the authors addressed the comment regarding the functionality of sRNA-cP, by performing CLIP-seq to profile small RNAs associated with AGO2, conducting UTR reporter assays with antisense oligos and small RNA mimics, and analyzing their effects on potential target mRNAs and cells.

However, these experiments are prone to nonspecific interactions and off-target/secondary effects. It is crucial to exclude these possibilities because the conclusions are inconsistent with many previous observations. For instance, it is well established that the AGO proteins become unstable when miRNA levels are reduced and that the Mid and PAZ domain of AGO interacts with the 5' phosphate and 3'-OH groups, respectively.

The authors conclude that the majority of AGO-associated sRNAs are not miRNAs and that the sRNA-cP (which are from abundant noncoding RNAs like rRNAs and snRNAs) are active regulators of mRNAs. If true, these findings will have a groundbreaking impact on our current understanding of the functions of noncoding RNAs and the regulation of mRNAs.

The following experiments would ensure the validity of the conclusions.

Firstly, the binding of sRNA-cP to AGO2 should be tested rigorously by using more stringent conditions to exclude any contaminations in the IP. And Northern blotting should be performed after AGO IP. It is important to include synthetic RNAs with the same sequences and known quantities loaded side-by-side as controls. This will allow absolute quantification of the sRNAs in AGO2 in comparison with miRNAs.

Secondly, to verify that AGO is the protein that indeed mediates the suppression of the UTR reporters, it is necessary to demonstrate the AGO-dependent suppression in a more direct manner. One can design the complementary target and conduct 5' RACE experiment to show that the site-specific cleavage occurs between nucleotides 10 and 11. In vitro experiments would also provide clear results. One can immunoprecipitated AGO2 and incubate with end-labeled target RNA with a complementary site to see if the cleavage occurs at the expected position.

Reviewer #3 (Remarks to the Author):

The revised manuscript is significantly improved and the new data supports many of the conclusions. Nonetheless, the manuscript still suffers from several weaknesses:

- The authors need to provide mass spectrometry data both as required by Nature journals and as a means for readers to judge the credibility of the results and conclusions. Where are the data for the spike-in standards or calibration curve for the qualitative and quantitative LC-MS studies? The authors do not present any of the data used to generate Extended Data Figure 1f and 1g. They cite a paper published in 2013 as the source of calibration curves but do not provide the data used in the current experiments. It should be obvious that calibration curves from 9 years ago would be entirely unacceptable for the current analyses, given changes in mass spectrometers with time and use, and that calibration curves must be prepared for every experiment. The authors also do not provide the raw or processed LC-MS data for the analyses or provide information about accessing the data in a public repository. These data must be provided for rigorous review.
- The manuscript remains very poorly written to the point of distraction. English language editing will be required improve the rigor of the manuscript and allow the reader to fully understand the work and assess the rigor of the authors' data and conclusions. For example, the statement on line 14, "...sRNA-OHs only possess about 10% of 15-30 nt mammalian sRNAs", which led to confusion during the review, is still extremely awkward. The authors appear to mean that the family of sRNAs consists of about 10% sRNA-OHs, so the phrase should reason something like: "...sRNA-OHs comprise about 10% of 15-30 nt mammalian sRNAs." This is not just being picky about language. The poor writing has led to significant confusion during the review process and will certainly affect readers.
- The legend for Extended Data Figure 1 is not consistent with the images. Panels a-d are fine. Panel e is noted to show LC-MS analyses but these appear to be shown in Panel g; Panel e shows two gels. Other panels are not consistent with the legend.

Point-by-point Response

Reviewer #1 (Remarks to the Author):

The authors adequately addressed most of my comments. I appreciate that. The paper has been very much improved by adding functional data for these RNAs.

However, the reply for this comment below is unsatisfactory.

“2. RNA modification interference: it is widely accepted that at least for tRNA fragments, the interfering modifications make it difficult for quantitative assessments. The authors mentioned this issue in the discussion, and proposes the demethylase treatment as an approach to increase accuracy. Not even trying this approach in this work is not compatible with the conclusions described in the just published work of PANDORA-seq in NCB.”

The authors tried something and gave up in their reply. AlkB treatment and sequencing has been used in many papers now. It is clear that extensive purification of the recombinant protein will be needed to remove contaminating ribonucleases from *E. coli*. For example, the authors can follow the AlkB protein purification steps (better yet use both AlkB and AlkB-D135S mutant, both plasmids can be obtained from Addgene) in this paper: PMID 26214130. Alternatively, they may contact Arraystar, inc. who may provide sequencing services using AlkB.

Response:

We really appreciate Reviewer #1's comments to try AlkB treatment to increase accuracy.

Fortunately, we obtained recombinant AlkB from Beyotime (R0639). We confirmed that the commercially available AlkB does have demethylase activity when we treated purified tRNA^{Gly}, tRNA^{His} and tRNA^{Val} from mouse liver total RNA with AlkB and detected m¹A abundance by LC-MS/MS (Extended Data Fig. 6a). In addition, no noticeable degradation of sRNAs was observed when treated with AlkB (Extended Data Fig. 6b).

According to Reviewer #1's comments, we treated mouse liver sRNAs with AlkB and then used for TANT-seq, and the method using AlkB-treated sRNAs for TANT-seq is termed ATANT-seq. As shown in Extended Data Fig. 6c-e, ATANT-seq reveals overall similar but slightly different sRNA expression pattern compared to TANT-seq. tsRNA-OH was slightly but significantly increased after AlkB treatment (Extended Data Fig. 6d), and the distribution of tsRNA-OH or tsRNA-cP from TNAT-seq and ATANT-seq

on a length scale was also similar but slightly different. These observations are similar with previous reports (*Nat Cell Biol* 2021, 23(4):424-436; *Cell Discov* 2021, 19;7(1):25).

The data for validation of AlkB activity and ATANT-seq are shown in Extended Data Fig. 6. The raw sequence data on ATANT-seq have been deposited in the NCBI Sequence Read Archive under BioProject number PRJNA725316. The related results are described on page 5 line 20-24 and page 6 line 1-6 of the revised manuscript with track changes, and the related methods are listed on page 36 line 5-10, page 39 line 26-29, page 40 line 1-7. In addition, the reference PMID 26214130 and other related references has been cited on page 5 line 22.

Reviewer #2 (Remarks to the Author):

In the revised manuscript, the authors addressed the comment regarding the functionality of sRNA-cP, by performing CLIP-seq to profile small RNAs associated with AGO2, conducting UTR reporter assays with antisense oligos and small RNA mimics, and analyzing their effects on potential target mRNAs and cells.

However, these experiments are prone to nonspecific interactions and off-target/secondary effects. It is crucial to exclude these possibilities because the conclusions are inconsistent with many previous observations. For instance, it is well established that the AGO proteins become unstable when miRNA levels are reduced and that the Mid and PAZ domain of AGO interacts with the 5' phosphate and 3'-OH groups, respectively.

Response:

We really appreciate Reviewer #2's comments for the potential non-specific interactions and off-target/secondary effects of sRNA-cPs.

To exclude these possibilities for non-specific interactions and off-target/secondary effects, we performed *in vitro* cleavage assay using Ago2. As shown in Fig. 7a-d, we used Myc-Ago2 immunoprecipitated from 293T cells co-transfected a plasmid expressing Myc-Ago2 and the indicated sRNA mimics to perform *in vitro* cleavage assay, and we found that Myc-Ago2 loaded with snR-2-cP, slncR-1-cP or smR-2-cP mimics significantly cleaved corresponding *Bcl2*, *Nr3C2* or *Twf1* 3' UTR fragments in a time-dependent manner. The 3' products from cleavage assay were cloned and sequenced, and the sequences of 3' products are exactly from the cleavage of target RNA between the nucleotides complementary to nucleotides 10 and 11 of the sRNA-cPs (Fig. 7b-d). Moreover, we found that Myc-Ago2 loaded with miR-122 mimic

specifically cleaved miR-122 target but not *Bcl2* 3' UTR fragment, and Myc-Ago2 loaded with snR-2-cP mimic specifically cleaved *Bcl2* 3' UTR fragment but not miR-122 target (Fig. 7e).

To further confirm whether sRNA-cPs can function in Ago2 complex as miRNAs to cleave target RNA, Myc-Ago2 was immunoprecipitated from 293T cells and mixed with sRNA mimics for *in vitro* cleavage assay. As shown in Fig. 7f-i, snR-2-cP, slncR-1-cP and smR-2-cP mimics guided the cleavage of *Bcl2*, *Nr3C2* and *Twf1* 3' UTR fragments respectively in a time-dependent manner as miR-122 to guide the cleavage of miR-122 target. These data show that sRNA-cPs can be loaded into Ago2 complex both *in vivo* and *in vitro* to guide the specific cleavage of target RNAs.

It has been concluded that “Stability of mammalian Ago2 depends on miRNA availability” by using a stable *Dicer1*^{-/-} MEF cell line to detect Ago2 or Ago1 protein level in the presence of protein synthesis inhibitor cycloheximide with or without transfection with siRNA duplex, miRNA duplex or constructs expressing pre-miRNA (*Nat Struct Mol Biol* 2013, 20(7):789-95). Therefore, the stability of mammalian Ago2 is likely dependent on the sRNA duplexes transfected or produced from *Dicer1*. Consistently, it has been reported that high Ago2 protein levels can be maintained by sRNA duplex instead of single-strand sRNAs in DGCR8 KO ESCs (*RNA* 2013, 19(5):605-12). In the presence of enough sRNA duplexes, it is likely that the Ago2 complex can keep working on cleavage of the target strand, and thus Ago2 is not necessary to be quickly degraded. When sRNA duplexes are not enough, it is not necessary to keep enough Ago2 to cleave the target RNAs, and thus Ago2 is reasonable to be quickly degraded.

It has been reported that the “5'-phosphate and 3'-hydroxyl ends are positioned in their respective binding pockets in the Mid and PAZ domains of Ago” (*Nature* 2008, 456(7224):921-6). It has been shown that Mid domain of human Ago2 can interact with 5'-P of sRNAs, and “the 5' base of the RNA stacks against Y529, which also forms a hydrogen bond to the 5' phosphate along with side chains of Y529, K533, N545, and K566” (*Science* 2012, 336(6084):1037-40). Currently, we still didn't find clear evidence from literatures whether PAZ domain directly interacts with 3'-OH group of sRNAs. Whether PAZ domain interacts with 3'-OH group might depend on the length of miRNA, since the length of miRNA is in the range of about 18-25 nt (*Science* 2005, 309(5740):1519-24; *Proc Natl Acad Sci U S A* 2009, 106(17):7016-21). In addition, isomirs are mature-miRNA variants that differ in length, sequence or both, and 3' isomirs such as 21-mer 3' isomir of miR-122 has weaker effect than longer miR-122 isomirs (*Nat Rev Mol Cell Biol* 2019, 20(1):21-37; *Nucleic Acids Res* 2017, 45(8):4743-

4755). Similarly, whether PAZ domain interacts with 3'-OH group of different 3' isomers is still unclear. Interestingly, miRNAs with 3' end biotinylation are still able to guide the cleavage activity of Ago2 (*Cell* 2015, 162(1):96-107), which indicates that 3'-OH is not required for sRNAs to guide the cleavage activity of Ago2.

To further confirm the effect of sRNA-cP and sRNA-OH, single-strand snR-2 and smR-2 with 5'-OH, 5'-P, 3'-OH or 3'-cP were prepared for cleavage assay. As shown in Fig. 7j, Ago2 complex guided by OH-snR-2-OH and OH-snR-2-cP, or P-snR-2-OH and P-snR-2-cP have similar cleavage activity for *Bcl2* 3' UTR fragment, which suggests that the existence of 3'-cP has no significant effect on the function of sRNA-cP in Ago2 complex. Similarly, 3'-cP is also not required for smR-2-cP to guide the cleavage of *Twf1* 3' UTR fragment in Ago2 complex (Fig. 7k). However, 5'-P is required for sRNA-cP to sufficiently function in Ago2 complex to cleave target RNA (Fig. 7j and k), which is consistent with the role of 5'-P in miRNA as described previously (*Science* 2012, 336(6084):1037-40; *Cell* 2012, 150(5):883-94).

The sequences for sRNAs and sRNA targets are shown in Supplementary Table 1. The data for *in vitro* cleavage assay are shown in Fig. 7. The related results are described on page 12 line 2-24 and page 13 line 1-13 of the revised manuscript with track changes. The related methods are listed on page 43 line 26-31, page 44 line 1-30 and page 45 line 1-16.

The authors conclude that the majority of AGO-associated sRNAs are not miRNAs and that the sRNA-cP (which are from abundant noncoding RNAs like rRNAs and snRNAs) are active regulators of mRNAs. If true, these findings will have a groundbreaking impact on our current understanding of the functions of noncoding RNAs and the regulation of mRNAs.

The following experiments would ensure the validity of the conclusions.

Firstly, the binding of sRNA-cP to AGO2 should be tested rigorously by using more stringent conditions to exclude any contaminations in the IP. And Northern blotting should be performed after AGO IP. It is important to include synthetic RNAs with the same sequences and known quantities loaded side-by-side as controls. This will allow absolute quantification of the sRNAs in AGO2 in comparison with miRNAs.

Response:

According to Reviewer #2's suggestion to exclude any contaminations in the IP, we showed the images for electrophoresis of sRNAs extracted from immunoprecipitation

by anti-Ago2 antibody and the control IgG in the revised manuscript. As shown in Fig. 6b, no detectable sRNAs were immunoprecipitated by control IgG, but obvious sRNAs extracted from Ago2 complex were detected.

According to Reviewer #2's suggestion, we performed Northern blotting after AGO IP. As shown in Fig. 6f, the presence of endogenous slncR-1-cP, snR-2-cP and smR-2-cP in extracted sRNAs or immunoprecipitated Ago2 complex from Hepa 1-6 cells was confirmed by Northern blot, and no sRNAs were detected when immunoprecipitated by control IgG. In addition, miR-21a were detected by Northern blot as a control (Fig. 6f).

According to Reviewer #2's suggestion, we also used synthetic miR-21a, slncR-1-cP, snR-2-cP and smR-2-cP detected by Northern blot at various concentrations to establish standard curves for each specific sRNA for absolute quantification of sRNAs, and the data are shown in Extended Data Fig. 11g.

The sequences for Northern blot probes are shown in Supplementary Table 1. The data for Northern blot analysis and quantification of sRNAs immunoprecipitated by Anti-Ago2 antibody from Hepa 1-6 cells are shown in Fig. 6f and Extended Data Fig. 10g. The related results are described on page 11 line 14-16 of the revised manuscript with track changes. The related methods are listed on page 40 line 10-21.

Secondly, to verify that AGO is the protein that indeed mediates the suppression of the UTR reporters, it is necessary to demonstrate the AGO-dependent suppression in a more direct manner. One can design the complementary target and conduct 5' RACE experiment to show that the site-specific cleavage occurs between nucleotides 10 and 11. *In vitro* experiments would also provide clear results. One can immunoprecipitated AGO2 and incubate with end-labeled target RNA with a complementary site to see if the cleavage occurs at the expected position.

Response:

We really appreciate Reviewer #2's suggestion to perform *in vitro* experiments using immunoprecipitated Ago2 incubated with end-labeled target RNA with a complementary site to see if the cleavage occurs at the expected position.

According to Reviewer #2's suggestion, we performed *in vitro* cleavage assay using Ago2. As shown in Fig. 7a-d, we used Myc-Ago2 immunoprecipitated from 293T cells co-transfected a plasmid expressing Myc-Ago2 and the indicated sRNA mimics to perform *in vitro* cleavage assays, and we found that Myc-Ago2 loaded with snR-2-cP,

slncR-1-cP or smR-2-cP mimics significantly cleaved corresponding *Bcl2*, *Nr3C2* or *Twf1* 3' UTR fragments in a time-dependent manner. The 3' products from cleavage assay were cloned and sequenced, and the sequences of 3' products are exactly from the cleavage of target RNA between the nucleotides complementary to nucleotides 10 and 11 of the sRNA-cPs (Fig. 7b-d). Moreover, we found that Myc-Ago2 loaded with miR-122 mimic specifically cleaved miR-122 target but not *Bcl2* 3' UTR fragment, and Myc-Ago2 loaded with snR-2-cP mimic specifically cleaved *Bcl2* 3' UTR fragment but not miR-122 target (Fig. 7e).

To further confirm whether sRNA-cPs can function in Ago2 complex as miRNAs to cleave target RNA, Myc-Ago2 was immunoprecipitated from 293T cells and mixed with sRNA mimics for *in vitro* cleavage assay. As shown in Fig. 7f-i, snR-2-cP, slncR-1-cP and smR-2-cP mimics guided the cleavage of *Bcl2*, *Nr3C2* and *Twf1* 3' UTR fragments respectively in a time dependent manner as miR-122 to guide the cleavage of miR-122 target. These data show that sRNA-cPs can be loaded into Ago2 complex both *in vivo* and *in vitro* to guide the specific cleavage of target RNAs.

The sequences for sRNAs, sRNA targets and Northern blot probes are shown in Supplementary Table 1. The data for Northern blot analysis and quantification of sRNAs immunoprecipitated by Anti-Ago2 antibody from Hepa 1-6 cells are shown in Fig. 6f and Extended Data Fig. 10g. The data for *in vitro* cleavage assay are shown in Fig. 7. The related results are described on page 11 line 14-16, page 12 line 2-24 and page 13 line 1-13 of the revised manuscript with track changes. The related methods are listed on page 40 line 10-21, page 43 line 26-31, page 44 line 1-30 and page 45 line 1-16.

Reviewer #3 (Remarks to the Author):

The revised manuscript is significantly improved and the new data supports many of the conclusions. Nonetheless, the manuscript still suffers from several weaknesses:

- The authors need to provide mass spectrometry data both as required by Nature journals and as a means for readers to judge the credibility of the results and conclusions. Where are the data for the spike-in standards or calibration curve for the qualitative and quantitative LC-MS studies? The authors do not present any of the data used to generate Extended Data Figure 1f and 1g. They cite a paper published in 2013 as the source of calibration curves but do not provide the data used in the current experiments. It should be obvious that calibration curves from 9 years ago would be entirely unacceptable for the current analyses, given changes in mass spectrometers

with time and use, and that calibration curves must be prepared for every experiment. The authors also do not provide the raw or processed LC-MS data for the analyses or provide information about accessing the data in a public repository. These data must be provided for rigorous review.

Response:

We really appreciate Reviewer #3's comments. According to Reviewer #3's comments, we deposited the raw unprocessed LC-MS/MS data in a public repository Figshare (<https://doi.org/10.6084/m9.figshare.23530554.v1>). We also deposited the data for the spike-in standard G (¹³C, ¹⁵N) and calibration curve data in Figshare at the above website. The calibration curves for RNA modification used in this study were freshly prepared for current analyses. According to Reviewer #3's suggestion, we showed calibration curves used in this study in Supplementary Table 2.

The calibration curves for LC-MS/MS are shown in Supplementary Table 2. The raw LC-MS/MS data have been deposited in Figshare (<https://doi.org/10.6084/m9.figshare.23530554.v1>). The related methods are listed on page 33 line 16-21 of the revised manuscript with track changes.

- The manuscript remains very poorly written to the point of distraction. English language editing will be required improve the rigor of the manuscript and allow the reader to fully understand the work and assess the rigor of the authors' data and conclusions. For example, the statement on line 14, "...sRNA-OHs only possess about 10% of 15-30 nt mammalian sRNAs", which led to confusion during the review, is still extremely awkward. The authors appear to mean that the family of sRNAs consists of about 10% sRNA-OHs, so the phrase should reason something like: "...sRNA-OHs comprise about 10% of 15-30 nt mammalian sRNAs." This is not just being picky about language. The poor writing has led to significant confusion during the review process and will certainly affect readers.

Response:

We really appreciate Reviewer #3's comments for English language editing.

According to Reviewer #3's comments, we replaced "sRNA-OHs only possess about 10% of 15-30 nt mammalian sRNAs" with "only about 10% of 15-30 nt mammalian sRNAs is sRNA-OHs" on page 13 line 17 of the revised manuscript with track changes.

According to Reviewer #3's comments, the following descriptions were corrected or modified.

On page 14 line 12-14 of the revised manuscript with track changes, "Interestingly, we found that miRNA only possesses about 10.5% of sRNA-OHs in mouse liver, which is consistent with the report that miRNA possesses about 16.6% of sRNA-OHs in mouse brain by SMARTer smRNA-Seq" was replaced with "Interestingly, we found that only about 10.5% of sRNA-OHs in mouse liver is miRNA, which is consistent with the report that about 16.6% of sRNA-OHs in mouse brain is miRNA by SMARTer smRNA-Seq".

On page 4 line 4 of the revised manuscript with track changes, "about 90% of 15-30 nt mammalian sRNAs are sRNA-cPs and sRNA-OHs" was replaced with "about 90% of 15-30 nt mammalian sRNAs can be categorized as either sRNA-cPs or sRNA-OHs".

On page 4 line 16 of the revised manuscript with track changes, "The TANT-seq data were highly reproducible" was replaced with "Additionally, the TANT-seq data demonstrated high reproducibility".

On page 4 line 17-20 of the revised manuscript with track changes, "the relative abundance of sRNA-OHs and sRNA-cPs is about 11-13% and 87-89% respectively in mouse liver, Hepa 1-6 and Hep G2 cells by combination of TANT-seq data and the ligation efficiency" was replaced with "By combining the TANT-seq data with ligation efficiency, the relative abundance of sRNA-OHs and sRNA-cPs is about 11-13% and 87-89% respectively in mouse liver, Hepa 1-6 and Hep G2 cells".

- The legend for Extended Data Figure 1 is not consistent with the images. Panels a-d are fine. Panel e is noted to show LC-MS analyses but these appear to be shown in Panel g; Panel e shows two gels. Other panels are not consistent with the legend.

Response:

We really appreciate Reviewer #3's careful reading to figure out the missing parts of the legend for Extended Data Fig. 1. The description for Extended Data Fig. 1e and 1f were added in revised manuscript, and the consistence of the panels with the legends was corrected.

REVIEWERS' COMMENTS

Reviewer #1 (Remarks to the Author):

The authors adequately addressed my comments.

Reviewer #2 (Remarks to the Author):

The authors have adequately addressed my comments, and the revised manuscript has strong evidence supporting the conclusions. I recommend publication of this ms.

Reviewer #3 (Remarks to the Author):

The authors have responded conscientiously to all of the reviewer comments. The manuscript is significantly improved and will be of interest to the community.